# A VAE for Transformers with Nonparametric Variational Information Bottleneck

**James Henderson**
Idiap Research Institute, Switzerland
`james.henderson@idiap.ch`

**Fabio Fehr**
Idiap Research Institute *and* EPFL, Switzerland
`fabio.fehr@idiap.ch`

## Abstract

We propose a Variational AutoEncoder (VAE) for Transformers by developing a Variational Information Bottleneck (VIB) regulariser for Transformer embeddings. We formalise such attention-based representations as mixture distributions, and use Bayesian nonparametrics to develop a Nonparametric VIB (NVIB) for them. The variable number of mixture components supported by nonparametrics captures the variable number of vectors supported by attention, and exchangeable distributions from nonparametrics capture the permutation invariance of attention. Our Transformer VAE (NVAE) uses NVIB to regularise the information passing from the Transformer encoder to the Transformer decoder. Evaluations of a NVAE, trained on natural language text, demonstrate that NVIB can regularise the number of mixture components in the induced embedding whilst maintaining generation quality and reconstruction capacity.

## 1 Introduction

Attention-based deep learning models, such as Transformers (Vaswani et al., 2017; Devlin et al., 2019), have achieved unprecedented empirical success in a wide range of cognitive tasks, in particular in natural language processing. The use of attention allows these models to represent their input with multiple vectors, which is essential for embedding natural language text (Bahdanau et al., 2015). On the other hand, deep variational Bayesian approaches to representation learning, such as variational autoencoders (VAEs) (Kingma & Welling, 2014), have also been shown to have many benefits (Mathieu et al., 2019; Ghosh et al., 2020; Vahdat & Kautz, 2020), especially due to their variational information bottleneck (VIB) (Alemi et al., 2017) for regularising the induced latent representations. However, it has not been clear how to combine these two trends, because the latent space induced by Transformers is a set of vectors whose size grows with the size of the input, whereas standard VIB methods only apply to a vector space of a fixed size (Liu & Liu, 2019; Fang et al., 2021; Park & Lee, 2021). To define a VIB regulariser for a Transformer's embedding space, we need to allow the size of a latent representation to vary dynamically depending on the complexity of the individual input, and yet regularise the total amount of information conveyed by the whole representation.

In this paper, we propose such a variational information bottleneck for variable sized latent representations, which we use to regularise the embeddings of a Transformer encoder-decoder, giving us a variational autoencoder for Transformers.[1] Like a Transformer encoder's embedding space, the proposed VAE's sampled encoder output is (a generalisation of) a set of vectors, and the decoder accesses this embedding with (a generalisation of) cross attention. But unlike Transformers, the proposed VIB layer for this VAE regularises the (effective) number of vectors in the set, as well as the information conveyed by each vector. We show that this regularisation improves generative abilities and compresses latent representations. In addition to the regularisation of over-parameterised language models (Child et al., 2019), previous work shows the efficacy of VAEs for: disentanglement (Higgins et al., 2017), language generation (Liu & Liu, 2019), and explainability (Mercatali & Freitas, 2021). All these topics are important and active areas of research in NLP.

To define this VIB, we need to model distributions over these variable-sized encoder embeddings, as interpreted by cross attention. Firstly, because the attention function returns an interpolation between the vectors output by the encoder, it generalises across the varying number of vectors, which like the input length is theoretically unbounded. Thus, to define distributions over these unbounded embeddings, we need to use nonparametric methods (Jordan, 2010). Secondly, the attention function is insensitive to the

---

[1]The code is available at https://github.com/idiap/nvib and https://github.com/idiap/nvib_transformers.

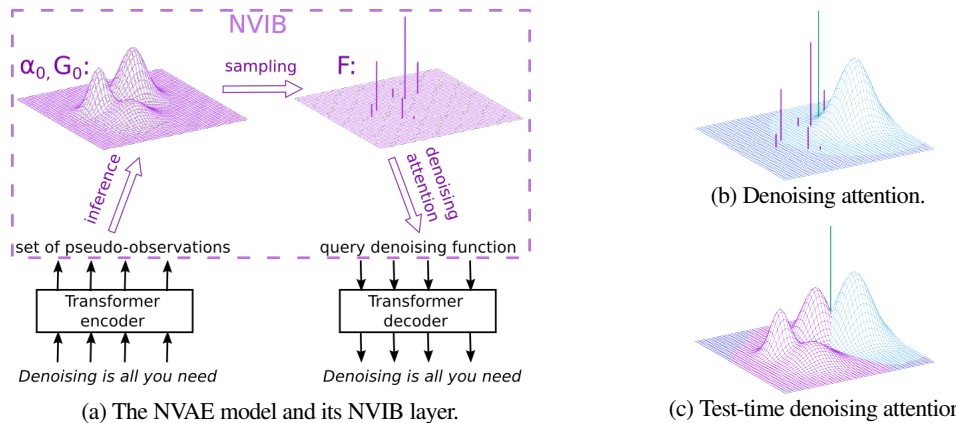

(a) The NVAE model and its NVIB layer.

(b) Denoising attention.

(c) Test-time denoising attention.

Figure 1: (a) Illustration of the NVAE model, with its NVIB layer. (b) Query denoising attention at training time, with the sampled distribution as the query prior, a noisy query observation, and the expected value of the denoised query. (c) Query denoising attention at test time using the mean distribution.

order of the vectors output by the encoder, so it interprets this embedding as a permutation-invariant set of vectors. Thus, the distributions over these permutation-invariant embeddings should be exchangeable (Jordan, 2010). Thirdly, the attention function imposes a normalised weighting over the embedding vectors, via the attention weights. So we should model an embedding as a distribution rather than a set.

A normalised weighting over an unbounded permutation-invariant set of fixed-length vectors matches exactly the properties of a nonparametric space of mixture distributions, which have been extensively studied in Bayesian nonparametrics using exchangeable distributions (Blei & Jordan, 2006; Jordan, 2010). In previous work, Bayesian nonparametrics is typically applied to learning *models* where the number of parameters grows with the size of the *training data* (Teh, 2010; Jordan, 2010; Kossen et al., 2021). In contrast, we apply it to inferring *latent representations* where the number of parameters grows with the size of the *input*. We believe this is the first work to use nonparametric methods in this way for deep variational Bayesian models.

To define a precise equivalence between attention-based representations and mixture distributions, we provide an interpretation of attention where the input set of vectors defines a mixture of impulse distributions, which is used as a prior to denoise the observed query vector (depicted in Figure 1b). Generalising sets of vectors to mixture distributions and generalising the attention function to query denoising allows us to propose a general deep variational Bayesian framework for attention-based models using Bayesian nonparametrics. More specifically, we propose to use Dirichlet processes (DPs) as the exchangeable distributions (Aldous, 1985; Jordan, 2010) to specify distributions over mixtures of impulse distributions, including distributions over the effective number of components in the mixture.

We define a nonparametric VIB (NVIB) layer using a bounded DP prior and posterior to regularise the effective size of variable-sized latent representations. This NVIB layer uses exact inference to infer the posterior from a set of pseudo-observations, and uses proposed efficient approximations to sample from this posterior with a reparameterisation trick and to regularise it with the KL divergence with the prior. Applying this NVIB regulariser to a Transformer autoencoder gives us our proposed nonparametric variational autoencoder (NVAE), depicted in Figure 1a. The noise introduced by sampling from the DP posterior controls the amount of information which flows from the encoder to the decoder, despite the fact that the amount of information required to reconstruct different text inputs varies enormously.

To evaluate the effectiveness of NVIB, we train a NVAE on natural language text and find that it is able to reconstruct, generate and regularise the effective number of vectors in the latent representation, thereby demonstrating that NVAE is a viable VAE. We also find that the regularised latent space is smooth, using a proposed method for interpolating between DP posteriors to generate interpolations between sentences.

**Contributions** This paper makes the following contributions: (1) We propose a variational Bayesian framework for modelling attention-based representations using mixture distributions, denoising attention and Bayesian nonparametrics (Section 2). (2) We propose a nonparametric variational information bottleneck (NVIB) regulariser for learning attention-based representations (Section 3). (3) We propose a nonparametric variational autoencoder (NVAE), which is a variational Bayesian extension of a Transformer encoder-decoder (Section 4). (4) We show that the NVAE model is a competitive VAE which can reconstruct, generate, regularise its latent space and intuitively interpolate between sentences (Section 5).

**Related work** Related work in stochastic attention assume that the keys, queries, values (Martin et al., 2020) or attention weight vectors of the network are treated as latent random variables (Deng et al., 2018; Bahuleyan et al., 2018; Fan et al., 2020; Cinquin et al., 2022). Nguyen et al. (2022) provides a formulation and interpretation of attention keys as latent mixture distributions, whereas our formulation characterises the whole attention function and is interpreted as Bayesian query denoising. The use of Bayesian nonparametrics to learn a variable sized latent space using a VAE (Nalisnick & Smyth, 2017; Goyal et al., 2017; Echraibi et al., 2020) still assumes a fixed-sized latent representation at test time, unlike our proposal.

## 2 A Nonparametric Bayesian Framework for Transformer Embeddings

This section proposes a formalisation of attention-based representations as mixture distributions over a vector space, and proposes nonparametric Bayesian methods for modelling information about these mixture distributions. First we show that standard attention functions can be interpreted as implementing Bayesian query denoising, where the set of vectors being accessed specifies a mixture of impulse distributions (Section 2.1). We adopt mixture distributions over vectors as a generalisation of attention-based representations, and adopt this denoising function as a generalisation of attention. Then we use Bayesian nonparametrics to propose prior (Section 2.2) and posterior distributions (Section 2.3) over these mixture distributions. These priors and posteriors form the basis of our nonparametric variational information bottleneck, proposed in Section 3.

### 2.1 Denoising Attention

The attention function provides access to a set of vectors by mapping a query vector to the resulting attention vector. As the basis of our approach to attention-based representations, we generalise the set of vectors to a probability distribution over vectors, and generalise attention to a function of these probability distributions.

The attention mechanism we assume is scaled dot product attention, standardly used in many attention-based models including Transformers. For simplicity, we consider cross attention, where a single query vector is mapped to a single result vector. This attention function projects the input vector $\boldsymbol{u}' \in \mathbb{R}^{1 \times p}$ via the weight matrix $\boldsymbol{W}^Q \in \mathbb{R}^{p \times d}$ to a query, and projects the set of vectors $\boldsymbol{Z} \in \mathbb{R}^{n \times p}$ via weight matrices $\boldsymbol{W}^K, \boldsymbol{W}^V \in \mathbb{R}^{p \times d}$ to keys and values, respectively. It uses the keys' dimensionality $d$ for scaling. We regroup this scaled dot product attention function into a core dot product attention function $\mathrm{Attn}(\boldsymbol{u}, \boldsymbol{Z})$ in which all operations are done in the space of $\boldsymbol{Z}$.

$$\mathrm{Attention}(\boldsymbol{u}', \boldsymbol{Z}; \boldsymbol{W}^Q, \boldsymbol{W}^K, \boldsymbol{W}^V) \;=\; \mathrm{Attn}(\boldsymbol{u}' \boldsymbol{W}^Q (\boldsymbol{W}^K)^T, \boldsymbol{Z}) \, \boldsymbol{W}^V \;=\; \mathrm{Attn}(\boldsymbol{u}, \boldsymbol{Z}) \, \boldsymbol{W}^V$$

where $\boldsymbol{u} = (\boldsymbol{u}' \boldsymbol{W}^Q (\boldsymbol{W}^K)^T) \in \mathbb{R}^{1 \times p}$. The function $\mathrm{Attn}(\boldsymbol{u}, \boldsymbol{Z})$ can then be defined in two equivalent ways (as shown in Appendix G): in terms of a sum over the vectors $\boldsymbol{z}_i$ in $\boldsymbol{Z}$, or in terms of an integral over a distribution which is only nonzero at the $\boldsymbol{z}_i$:

$$\mathrm{Attn}(\boldsymbol{u}, \boldsymbol{Z}) \;=\; \mathrm{softmax}\left(\tfrac{1}{\sqrt{d}} \boldsymbol{u} \boldsymbol{Z}^T\right) \boldsymbol{Z} \;=\; \mathrm{DAttn}(\boldsymbol{u}; F_{\boldsymbol{Z}}) \tag{1}$$

$$F_{\boldsymbol{Z}} \;=\; \sum_{i=1}^{n} \frac{\exp(\frac{1}{2\sqrt{d}}||\boldsymbol{z}_i||^2)}{\sum_{i=1}^{n}\exp(\frac{1}{2\sqrt{d}}||\boldsymbol{z}_i||^2)} \, \delta_{\boldsymbol{z}_i} \tag{2}$$

$$\mathrm{DAttn}(\boldsymbol{u}; F) \;=\; \int_{\boldsymbol{v}} \frac{f(\boldsymbol{v}) \, g(\boldsymbol{u}; \boldsymbol{v}, \sqrt{d}\boldsymbol{I})}{\int_{\boldsymbol{v}} f(\boldsymbol{v}) \, g(\boldsymbol{u}; \boldsymbol{v}, \sqrt{d}\boldsymbol{I}) \, d\boldsymbol{v}} \, \boldsymbol{v} \, d\boldsymbol{v} \tag{3}$$

where $\delta_{\boldsymbol{z}_i}$ is an impulse distribution at $\boldsymbol{z}_i$, $f(\cdot)$ is the probability density function for distribution $F$, and $g(\boldsymbol{u}; \boldsymbol{v}, \sqrt{d}\boldsymbol{I})$ is the multivariate Gaussian function with diagonal variance of $\sqrt{d}$. As depicted in Figure 1b, $\mathrm{DAttn}(\boldsymbol{u}; F_{\boldsymbol{Z}})$ can be interpreted as query denoising. The query $\boldsymbol{u}$ is interpreted as an observation of some true vector $\boldsymbol{v}$ which has been corrupted by Gaussian noise, where $\boldsymbol{v}$ was generated from a prior probability distribution $F_{\boldsymbol{Z}}$ specified by $\boldsymbol{Z}$. The result of $\mathrm{Attn}(\boldsymbol{u}, \boldsymbol{Z})$ is the expected value of this true vector $\boldsymbol{v}$ after seeing the noisy observation $\boldsymbol{u}$, which can be interpreted as a form of denoising.

This *denoising attention* function $\mathrm{DAttn}(\boldsymbol{u}; F)$ is actually a generalisation of attention over a set of vectors, in that it is defined for any probability distribution $F$ over a vector space. In the special case where $F = \sum_i \pi_i \delta_{\boldsymbol{z}_i}$ is a finite mixture of impulse distributions, it is the same as $\mathrm{Attn}(\boldsymbol{u}, \boldsymbol{Z})$ but with a bias term $\log(\pi_i)$ substituted for $\frac{1}{2\sqrt{d}}||\boldsymbol{z}_i||^2$. In the rest of this paper, we will use $\mathrm{DAttn}(\boldsymbol{u}; F)$ as our definition of attention, which allows us to treat the latent space of a Transformer encoder-decoder as mixture distributions.

## 2.2 A Prior over Mixture Distributions

Given that our attention-based latent representations are formalised as mixture distributions $F$, a Bayesian approach requires a prior over these distributions. Attention-based models place no finite bound on the possible number of vectors in their set of vectors $\boldsymbol{Z}$, and thus there is no finite bound on the number of parameters needed to specify the equivalent mixture distribution $F$. Nonetheless, we can still specify probability distributions over this infinite space of possible distributions $F$, using methods from Bayesian nonparametrics. These nonparametric Bayesian methods, with exchangeable distributions, are specifically designed for modelling probability distributions over unboundedly large mixture distributions.

We base our distributions over mixture distributions on Dirichlet processes $\text{DP}(G_0,\alpha_0)$. Dirichlet processes (DPs) are a generalisation of Dirichlet distributions to an infinite support, such as the points in a vector space. A Dirichlet distribution $\text{Dir}(\boldsymbol{\alpha})$ is a distribution over probability mass functions $\boldsymbol{\pi}$ of discrete categories $i$, $1 \leq i \leq \kappa$. One useful definition of Dirichlet processes views a DP $F \sim \text{DP}(G_0,\alpha_0)$, where $G_0$ is the base distribution over vectors and $\alpha_0 \in \mathbb{R}$ is the concentration parameter, as the limit of a sequence of finite Dirichlet distributions (see Teh (2010)), given in equation 4. Note that the Dirichlet distributions in equation 4 are symmetric, in that all the $\kappa$ categories $i$ have the same $\alpha_i = \frac{\alpha_0}{\kappa}$ parameter values. However, these categories end up with very different weights $\pi_i$, due to the most probable categories getting a large proportion of the probability mass and the tail of categories getting an exponentially decreasing amount of probability mass. In the infinite limit, this tail is infinitely long with infinitesimal probabilities. The number of categories which get nontrivial probabilities is determined by $\alpha_0$, and becomes independent of $\kappa$ as $\kappa$ gets large.

As shown in this definition, each sample $F$ from a DP is an infinite mixture of impulse distributions $\delta_{\boldsymbol{z}_i}$, parameterised by an infinite sequence of weight-vector pairs $\pi_i, \boldsymbol{z}_i$. This contrasts with the finite $\boldsymbol{Z}$ in attention-based representations. Having an infinite $F$ would also cause problems in our variational Bayesian model, because VIB uses a bound on the log-likelihood (see Section 3.1), which Kingma & Welling (2014) showed has an error of $D_{\text{KL}}(q(F \,|\, x) \,\|\, p(F \,|\, x))$ (the looseness of the bound). This would be infinite unless both the true posterior $p(F \,|\, x)$ and its approximation $q(F \,|\, x)$ generate a finite $F$, so we need a prior which generates finite $F$.

$$
\begin{aligned}
F &= \sum_{i=1}^{\infty} \pi_i \delta_{\boldsymbol{z}_i} \qquad\qquad (4) \\
\boldsymbol{\pi} &\sim \lim_{\kappa \to \infty} \text{Dir}(\tfrac{\alpha_0}{\kappa}, \overset{\kappa}{...}, \tfrac{\alpha_0}{\kappa}) \\
\boldsymbol{z}_i &\sim G_0 \quad \text{for } i=1,...,\infty
\end{aligned}
$$

$$
\begin{aligned}
F &= \sum_{i=1}^{\kappa_0} \pi_i \delta_{\boldsymbol{z}_i} \qquad\qquad (5) \\
\boldsymbol{\pi} &\sim \text{Dir}(\tfrac{\alpha_0}{\kappa_0}, \overset{\kappa_0}{...}, \tfrac{\alpha_0}{\kappa_0}) \\
\boldsymbol{z}_i &\sim G_0 \quad \text{for } i=1,...,\kappa_0
\end{aligned}
$$

**The Unbounded Dirichlet Process Prior** We do not want a prior which places an apriori bound on the size of $F$, so we assume it is finite but unbounded, and propose a prior which is an unbounded sequence of finite approximations to a DP. We define a bounded DP $F \sim \text{BDP}(G_0,\alpha_0,\kappa_0)$ as in equation 5. Our approach to the prior is to use an unbounded but finite $\kappa_0$, so we define a distribution over approximations as $\kappa_0$ increases towards infinity. Hence, every distribution is over a finite number of vectors, but there is no finite bound on the number of vectors in all distributions. Given $\phi$ is some distribution over positive integers $\kappa \in \mathbb{Z}^+$, we define this unbounded DP as $\text{UDP}(G_0,\alpha_0,\phi) = \text{BDP}(G_0,\alpha_0,\kappa)$ where $\kappa \sim \phi$.

We use these definitions both to define a general prior over probability distributions, and to define a conditional prior for each input length. In both cases, the base distribution $G_0^p$ is assumed to be a unit Gaussian (inspired by Kingma & Welling (2014)) and the concentration parameter $\alpha_0^p$ is assumed to be one.[2]

$$
G_0^p = \mathcal{N}(\boldsymbol{\mu}^p, \boldsymbol{I}(\boldsymbol{\sigma}^p)^2); \quad \alpha_0^p = 1; \quad \boldsymbol{\mu}^p = \boldsymbol{0}; \quad \boldsymbol{\sigma}^p = \boldsymbol{1}
$$

The general prior is $\text{UDP}(G_0^p, \alpha_0^p, \phi^p)$, where the size distribution $\phi^p$ is determined empirically. The conditional prior $\text{BDP}(G_0^p, \alpha_0^p, \kappa_0)$ sets the level of approximation $\kappa_0$ as a fixed function of the input length $n$, in particular $\kappa_0 = (n+1)\kappa^\Delta$, where $\kappa^\Delta \in \mathbb{Z}^+$ is a hyperparameter that controls the approximation.

**A Conditional Bounded DP Prior** It will be useful to generalise this conditioning for the level of approximation to any conditional prior which is a fixed function of only the input length. If we know the input length $n$, but know nothing about the content of the text, then the distribution of vectors should stay the same as the general prior, $G_0^{p'} = G_0^p$. However, the count of observations we expect to have after an input

---

[2]We will use "p" and "q" superscripts to designate variables for the prior and posterior, respectively. Similarly, a zero subscript is part of the name of the variable, in contrast to positive integer subscripts which are indices.

of that length would not be $\alpha_0^p$, but should include a pseudo-count $\alpha^\Delta \in \mathbb{R}_{\geq 0}$ hyperparameter for every token, and thus $\alpha_0^{p'} = \alpha_0^p + n\alpha^\Delta$. This then gives us the conditional prior given $n$ of $\mathrm{BDP}(G_0^p, \alpha_0^{p'}, \kappa_0)$.

## 2.3 A Posterior over Mixture Distributions

Since a DP is a conjugate prior, we can use exact inference to compute the posterior DP from the prior DP plus a set of pseudo-observations output by the encoder. Each pseudo-observation is a real-valued pseudo-count $\alpha_i^q \in \mathbb{R}_{\geq 0}$ and a parametric distribution which represents uncertainty in the observation. We use an isotropic Gaussian, $G_i^q = \mathcal{N}(\boldsymbol{\mu}_i^q, \boldsymbol{I}(\boldsymbol{\sigma}_i^q)^2)$, as the parametric distribution, specified by a mean $\boldsymbol{\mu}_i^q \in \mathbb{R}^{1 \times d}$ and a standard deviation $\boldsymbol{\sigma}_i^q \in \mathbb{R}_{>0}^{1 \times d}$. Here we assume that the number of candidate pseudo-observations is the same as the length $n$ of the input, but some of these pseudo-observations may have zero pseudo-counts and thus be effectively removed from the set. The formula for exact inference of the posterior DP is given in equation 6, where there is an $n+1^{\text{th}}$ component of the base distribution $G_0^q$ which comes from the prior, namely $\alpha_{n+1}^q = \alpha_0^p$ and $G_{n+1}^q = G_0^p$.

$$
\begin{aligned}
F &\sim \mathrm{DP}(G_0^q, \alpha_0^q) & (6) \\
\alpha_0^q &= \sum_{i=1}^{n+1} \alpha_i^q \; ; \quad G_0^q = \sum_{i=1}^{n+1} \frac{\alpha_i^q}{\alpha_0^q} G_i^q
\end{aligned}
$$

$$
\begin{aligned}
F &= \sum_{i=1}^{n+1} \rho_i F_i & (7) \\
\boldsymbol{\rho} &\sim \mathrm{Dir}(\alpha_1^q, ..., \alpha_{n+1}^q) \\
F_i &\sim \mathrm{BDP}(G_i^q, \alpha_i^q, \kappa^\Delta) \quad \text{for } i = 1, ..., n+1
\end{aligned}
$$

We derive an alternative factorisation of the posterior DP (Appendix H) which helps with the sampling method in Section 3.2. We then bound this factorised DP so that it generates the same number $\kappa_0 = (n+1)\kappa^\Delta$ of weighted vectors as the prior $\mathrm{BDP}(G_0^p, \alpha_0^p, \kappa_0)$. The resulting bounded posterior $F \sim \mathrm{BFDP}(\boldsymbol{G}^q, \boldsymbol{\alpha}^q, \kappa^\Delta)$ is given in equation 7, which defines our posterior distribution $q(F \mid x)$. This posterior simplifies to a mixture $F = \sum_{i=1}^{n+1} \sum_{j=1}^{\kappa^\Delta} \rho_i \pi_{ij}' \delta_{\boldsymbol{z}_{ij}}$ of impulse distributions, with $\boldsymbol{\rho} \sim \mathrm{Dir}(\alpha_1^q, ..., \alpha_{n+1}^q)$, $\boldsymbol{\pi}_i' \sim \mathrm{Dir}(\frac{\alpha_i^q}{\kappa^\Delta}, \overset{\kappa^\Delta}{...}, \frac{\alpha_i^q}{\kappa^\Delta})$, and $\boldsymbol{z}_{ij} \sim G_i^q$.

**The Mean Posterior Mixture Distribution** A VAE is trained on samples from the posterior, but at test time VAEs typically use the mean of this distribution. Generalising the latent space to mixture distributions makes this straightforward, since the mean of our BFDP posterior is its base distribution $G_0^q$. This base distribution is a continuous distribution, whereas at training time all samples are discrete distributions. Nonetheless, when accessed via denoising attention, the base distribution looks like a typical sample from the posterior. This is visualised in Figure 1 by comparing the vector returned by denoising attention (in green) given the continuous mean distribution (Figure 1c) and a typical sample from this distribution (Figure 1b). Thus, the function defined by applying denoising attention to a sampled distribution can be seen as a noisy version of the function defined by applying denoising attention to the mean distribution. For our Gaussian mixture base distribution $G_0^q$, there is a closed-form solution to computing denoising attention, given in Appendix F.

## 3 The Nonparametric Variational Information Bottleneck

By generalising attention-based representations to mixture distributions and generalising the attention function to denoising attention, we can define a VIB regulariser for attention-based interfaces. Such encoder-decoder interfaces take a set-of-vectors representation and return a function from query to result vectors. We map the input set-of-vectors to a set of pseudo-observations, and define the returned function with denoising attention. Then, given the nonparametric prior and posterior from Section 2, we can define our nonparametric VIB regulariser by specifying how to compute the KL divergence between the prior and posterior, and how to effectively sample from the posterior for training. As far as we are aware, this proposal is the first VIB model for attention-based representations like Transformer embeddings.

The VIB layer in a VAE controls the amount of information passing through it by introducing noise according to a posterior output by the encoder, and regularises this information by minimising the KL divergence between this posterior and an uninformative prior. One of the known difficulties with VAEs is that they can be difficult to train due to the posterior collapsing to the prior (Bowman et al., 2016). Similarly to the *free-bits* objective proposed to address this problem for vector-space VAEs (Kingma et al., 2016), instead of regularising towards the prior, which gets no pseudo-counts from the input, we regularise towards the conditional

prior $\mathrm{BDP}(G_0^p, \alpha_0^{p'}, \kappa_0)$, which gets $n\alpha^\Delta$ pseudo-counts from the input but knows nothing about the information they convey. We found that this helps with the stability of training and avoiding posterior collapse.

## 3.1 THE VARIATIONAL INFORMATION BOTTLENECK LOSS

The evidence lower bound (ELBO) is commonly used in variational Bayesian methods as an objective which approximately maximises the log-likelihood of the observation $x$, where $x$ is the input text.

$$\log(p(x)) \;\geq\; \mathbb{E}_{q(F|x)}\log(p(x|F)) - D_{\mathrm{KL}}(q(F|x) \,\|\, p(F)) \tag{8}$$

$$L_R \;=\; -\mathbb{E}_{q(F|x)}\log(p(x|F))$$

The first term of the bound is the reconstruction loss $L_R$, computed using samples $F$ from the approximate posterior $q(F|x)$, and the second term is the KL divergence between this posterior and the prior $p(F)$.

For the KL divergence term, since both the prior and the posterior are conditioned on the same bound $\kappa_0$ on the number of vectors they generate, we can compute a meaningful finite KL divergence between the conditional prior $p(F)=\mathrm{BDP}(G_0^p, \alpha_0^{p'}, \kappa_0)$ and the posterior $q(F|x)=\mathrm{BFDP}(\boldsymbol{G}^q, \boldsymbol{\alpha}^q, \kappa^\Delta)$, given $\kappa^\Delta$ and $\kappa_0=(n+1)\kappa^\Delta$. The derivation of the KL divergence is given in Appendix I. It has two terms, one $L_D$ for the distribution of weights $\boldsymbol{\pi}$ generated by the Dirichlet distributions, and one $L_G$ for the distribution of vectors $\boldsymbol{Z}$ generated by the component Gaussians. Using the exact KL divergence for our bounded DPs would regularise each component equally, even for components which have zero $\alpha_i^q$ and thus have no impact on the posterior. We instead approximate a KL divergence where only samples with a nontrivial weight are regularised. Marginalising over the number of nontrivial weights for each component does not appear to be tractable for $L_D$, but since the relationship is approximately linear (see Appendix I), we simply substitute the expected number $\frac{\alpha_i^q}{\alpha_0^q}\kappa_0$ for the actual number in the equation for KL divergence. This gives us the following loss terms for the KL divergence, where $\Gamma$ is the gamma function and $\psi$ is the digamma function.

$$L_D + L_G \;\approx\; D_{\mathrm{KL}}(q(F|x) \,\|\, p(F)) \tag{9}$$

$$L_D \;=\; \log\Gamma(\alpha_0^q) - \log\Gamma(\alpha_0^{p'}) + (\alpha_0^q - \alpha_0^{p'})\left(-\psi(\alpha_0^q) + \psi(\frac{\alpha_0^q}{\kappa_0})\right) + \kappa_0\left(\log\Gamma(\frac{\alpha_0^{p'}}{\kappa_0}) - \log\Gamma(\frac{\alpha_0^q}{\kappa_0})\right)$$

$$L_G \;=\; \tfrac{1}{2}\kappa_0 \sum_{i=1}^{n+1} \frac{\alpha_i^q}{\alpha_0^q} \sum_{h=1}^{d}\left(\frac{(\mu_{ih}^q - \mu_h^p)^2}{(\sigma_h^p)^2} + \frac{(\sigma_{ih}^q)^2}{(\sigma_h^p)^2} - 1 - \log\frac{(\sigma_{ih}^q)^2}{(\sigma_h^p)^2}\right)$$

Both $L_D$ and $L_G$ scale approximately linearly with $\kappa_0$.

To generalise the ELBO beyond autoencoders, it can be viewed as a way to regularise the amount of information which passes through the latent representation (Alemi et al., 2017). This VIB interpretation allows the different parts of the objective to have different weights. We introduce two hyperparameters to control the relative weight of the above three parts of the ELBO, which defines our VIB loss $L$.

$$L \;=\; L_R + \lambda_D L_D + \lambda_G L_G \tag{10}$$

## 3.2 SAMPLING A MIXTURE DISTRIBUTION FROM THE POSTERIOR

To control the amount of information which passes from the encoder to the decoder, at training time a VAE (Kingma & Welling, 2014) samples from the encoder's posterior distribution and uses this sample to reconstruct the input. The "reparameterisation trick" is used to ensure that backpropagation of the reconstruction error through this sampling step can be done effectively. We propose a novel reparameterisation trick for bounded Dirichlet processes which allows sampling without any categorical choices, and propose specific sampling methods which result in effective backpropagation through the sampling step.

For our NVIB model, we sample the parameters $\langle \boldsymbol{\pi}, \boldsymbol{Z} \rangle$ of a mixture distribution $F$ generated by our bounded Dirichlet process posterior $\mathrm{BFDP}(\boldsymbol{G}^q, \boldsymbol{\alpha}^q, \kappa^\Delta)$, where $F$ consists of a set of impulse distributions $\delta_{\boldsymbol{z}_i}$ each with a weight $\pi_i$. A straightforward approach to sampling from a Dirichlet process would independently sample weights $\boldsymbol{\pi}$ from a (theoretically infinite) Dirichlet distribution and sample vectors $\boldsymbol{Z}$ from the base distribution of the DP, where sampling from the base distribution involves first sampling a component of the base distribution and then sampling a vector from that component's Gaussian.

**A Factorised Sampling Method** The problem is that sampling a component is a discrete choice, for which there is no exact reparameterisation trick. Instead, we note that the components do not differ in the

number of vectors sampled from each one (always theoretically infinite for a DP), but only differ in the distribution of weights for those vectors. As specified in the factorised DP in Section 2.3, we characterise this distribution over weights by factorising it into two steps: first choosing how the total weight is distributed across components ($\rho$), and then for each component choosing how its weight is distributed across its vectors ($\pi'_i$). These are both continuous choices. The vectors can then be sampled independently from each component.

**Reparameterisation tricks**    Each individual component specifies a Gaussian distribution over vectors, so we can use the same reparameterisation trick as Kingma & Welling (2014) for sampling vectors $\boldsymbol{Z}_i$ from an individual component $i$.

The factorised and bounded nature of our posterior $\mathrm{BFDP}(\boldsymbol{G}^q,\boldsymbol{\alpha}^q,\kappa^\Delta)$ means that the total weights $\boldsymbol{\rho}$ and the individual weights $\boldsymbol{\pi}'_i$ are all sampled from Dirichlet distributions. A Dirichlet distribution over a set of category weights can be sampled by sampling from a Gamma distribution for each category and then normalising. There is no closed-form, explicit reparameterisation trick for the exact Gamma distribution, but there are for approximations. Knowles (2015) proposes two such approximations which we combine, one which is more accurate for small $\alpha$ and one which is more accurate for larger $\alpha$. We leave the investigation of implicit reparameterisation gradients (Figurnov et al., 2018), an alternative approach to explicit reparameterisation, to future work. More specifics about the sampling methods and their reparameterisation tricks is given in Appendix J. We provide an evaluation (Appendix B.3) showing that $\kappa^\Delta{=}1$ is an efficient and effective sampling method. In this case, there is no need to sample from the Dirichlet for each individual component, but we still sample the weights $\boldsymbol{\rho}$ across components.

## 4    THE NONPARAMETRIC VARIATIONAL AUTOENCODER

We define a VAE for Transformers by using the nonparametric VIB defined in Section 3 to regularise the attention-based representation between the encoder and decoder of a Transformer autoencoder, as depicted in Figure 1a. In this NVAE model, the Transformer encoder is used to estimate the parameters $\langle\boldsymbol{\alpha}^q,\boldsymbol{\mu}^q,\boldsymbol{\sigma}^q\rangle$ of the posterior given the input text $x$. The Transformer decoder is used to reconstruct the input text $x$ using denoising attention over a sample $F$ from this posterior.

**The NVIB Regulariser**    Our NVIB layer regularises the amount of information which passes from the encoder to the decoder through this posterior. As with VIB for vector spaces, the KL divergence encourages the encoder to output component Gaussians with smaller $\boldsymbol{\mu}^q_i$ and larger $\boldsymbol{\sigma}^q_i$. With NVIB, the KL divergence also encourages the encoder to output smaller and sparser $\alpha^q_i$, which regularises the effective number of components in the posterior as well as the noisiness of their weights.

**The Transformer Encoder**    The Transformer encoder $q(F|x)$ with a text $x$ of $n$ number of tokens is used to compute a vector for each token $i$ of the input. From each of these $n$ individual token embeddings, the encoder then linearly projects to three parameters, $\alpha^q_i \in \mathbb{R}$, $\boldsymbol{\mu}^q_i \in \mathbb{R}^{1\times p}$ and $\log(\boldsymbol{\sigma}^q_i) \in \mathbb{R}^{1\times p}$. The variance parameters are exponentiated to be strictly positive, whereas the pseudo-count parameters $\alpha^q_i$ are estimated using a Rectified Linear Unit (ReLU) activation (Nair & Hinton, 2010), which results in masking the vector during cross-attention when it is exactly zero. Thus, the DP posterior has one component $\langle\alpha^q_i,\boldsymbol{\mu}^q_i,\boldsymbol{\sigma}^q_i\rangle$ of its base distribution for each token of the input (plus one for the prior).

**The Transformer Decoder**    The Transformer decoder $q(x|F)$ receives a distribution $F$ over vectors and reconstructs the input text $x$. During training, $F$ is specified by the sampled vectors $\mathbf{Z}\in\mathbb{R}^{\kappa_0\times p}$ and the sampled weights $\boldsymbol{\pi}\in\mathbb{R}^{\kappa_0\times 1}$, and at test time $F$ is specified by the output of the encoder $\boldsymbol{\alpha}^q\in\mathbb{R}^{\kappa_0\times 1}$, $\boldsymbol{\mu}^q\in\mathbb{R}^{\kappa_0\times p}$ and $\log(\boldsymbol{\sigma})^q\in\mathbb{R}^{\kappa_0\times p}$. In both cases, the decoder accesses $F$ using denoising attention in the same way that standard Transformer decoders use cross attention. We include the exact equations used for a deep learning implementation of denoising attention in Appendix K. During training, the text is predicted using teacher forcing, and during test time the text is predicted autoregressively using greedy decoding until the end-of-sequence token is generated or the sentence generated is 50 tokens larger than the target length.

**The Generative Model**    To use our NVAE model as a generative model, we sample from the prior and use the trained Transformer decoder to generate a sentence. As discussed in Section 2.2, to sample from the same prior as used for training, we need to first sample a sentence length, and then sample from the conditional prior given that (approximate) sentence length. For simplicity, we sample the sentence length from the empirical distribution of sentence lengths in the training data.

# 5 INTRINSIC EVALUATIONS OF NVIB IN NVAE

To support our theoretical contributions, we provide proof-of-concept experiments which demonstrate that our proposed NVIB regulariser performs as claimed. We evaluate it in our proposed NVAE model by training NVAEs on natural language text and evaluating the resulting models. We show that the NVAE is a viable VAE model as it exhibits a competitive reconstruction versus generation trade-off (Section 5.1). We show that the NVIB layer is able to dynamically choose the number of components it needs in its embeddings (Section 5.2). Additionally, NVIB provides an intuitive way to interpolate between sentence embeddings, which provides an evaluation of the smoothness of the latent space (Section 5.3).

**Data** The Wikitext-2 and Wikitext-103 (Merity et al., 2017) encyclopedia datasets were selected as they are general English language corpora of a small and large scale containing high quality Wikipedia articles.

**Baselines** We compare to various alternative ways to define a VAE from a Transformer autoencoder. As representative of a standard fixed-length-vector VAE, the Variational Transformer Pooled (VTP) baseline pools its vectors across the sequence length dimension, and then applies a Gaussian VIB layer (Kingma & Welling, 2014). At the other extreme, the Variational Transformer (VT) baseline keeps all its vectors and applies a Gaussian VIB layer to each one. In between these baselines, as a hand-coded solution to constraining the quantity of latent vectors, Variational Transformer Stride (VTS) baselines, with parameter $S$, masks $1-S$ proportion of the embedding vectors based on their position. For comparability, all our baselines only differ from the NVAE model in the latent representation between the encoder and decoder, with the same Transformer encoder and Transformer decoder architectures.

## 5.1 RECONSTRUCTION VERSUS GENERATION

This section shows that the NVAE Transformer model is a competitive VAE in both reconstruction of input sentences and generation by sampling from the prior. All models undergo hyperparameter tuning on the validation set (Appendix B) across 5 seeds, to select the best models and then report results on the Wikitext-2 test set. For reconstruction, we report the SacreBLEU metric (Papineni et al., 2002; Post, 2018). For generation, we report forward perplexity (F-PPL) and reverse perplexity (R-PPL) (Zhao et al., 2018; Cífka et al., 2018), which trains an external language model on the gold training text and evaluates it on the generated text (F-PPL) or vice versa (R-PPL). Training details are provided in Appendix A.

Figures 2a and 2b show the reconstruction versus generation trade-off on the Wikitext-2 test set, where lower right is better. The single vector baseline VTP is unable to reconstruct well (low BLEU) or generate diverse sentences (high R-PPL). Even with larger capacity and more training data, the model performs poorly in generation (Appendix B.4). The full vector baseline VT is unable to consistently generate fluent sentences (high F-PPL), whereas the position-based dropout baseline VTS shows that regularisation of the space is beneficial for both reconstruction and generation quality.

The best NVAE models are competitive with the best VTS baselines in being able to both reconstruct and generate, even slightly better at generation with high reconstruction accuracy. In particular, reverse perplexity shows that NVAEs are able to generate a diverse collection of sentences from the prior, and forward perplexity shows that these are fluent sentences (samples in Appendix C). We believe this advantage is the result of learning what vectors to keep instead of a hand-coded position-based dropout.

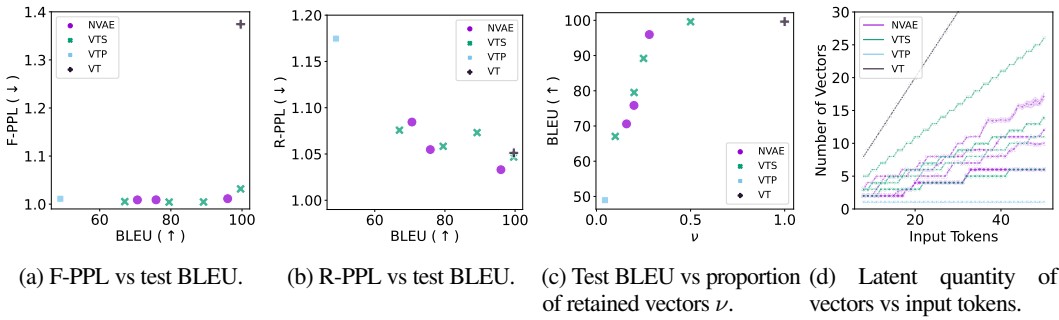

(a) F-PPL vs test BLEU.    (b) R-PPL vs test BLEU.    (c) Test BLEU vs proportion of retained vectors $\nu$.    (d) Latent quantity of vectors vs input tokens.

Figure 2: Reconstruction versus generation trade-off (a), (b) and regularisation analysis (c), (d).

## 5.2 REGULARISATION

This section shows that the NVIB layer is able to regularise the number of vectors in the latent representation of a NVAE. Without regularisation the NVIB layer becomes a standard transformer without any noise and retaining all latent vectors (ablation in Appendix B.1). The proportion of vectors retained is controllable by the conditional prior hyperparameter $\alpha^\Delta$. Figure 2c shows that there exist $\alpha^\Delta$ where the NVAE is able to remove a significant proportion $(1-\nu)$ of vectors whilst maintaining a high reconstruction performance. Figure 2d plots the number of latent vectors retained during evaluation against the number of input tokens. The NVAE models are able to learn to dynamically regularise the number of vectors based on the information within the text, without hand-coding a function of length as in VTS. The different NVAE lines show the effect of $\alpha^\Delta$ on vector retention, with larger values allowing more vectors to be used (also see the evaluation in Appendix B.2).

## 5.3 INTERPOLATION

The NVIB framework provides an intuitive interpolation between latent sets of vectors that overcomes the challenges with the baselines of latent vector alignment (analysis in Appendix D) and variable set sizes. We simply interpolate the probability assigned to each point in vector space. Given two latent mixture distributions $F_1$ and $F_2$, we decode from the combined mixture distribution $(\tau F_1 + (1-\tau)F_2)$, for varying interpolation rates $0 \leq \tau \leq 1$. For the baselines we use $\tau \mathbf{Z}_1 + (1-\tau)\mathbf{Z}_2$ such that the interpolation is over the content of the latent vectors $\mathbf{Z}_i$. We align latent sets of vectors by input position and pad smaller sets with zero vectors, which is the mean of the Gaussian prior. We evaluate the interpolation with selected larger scale models (see Appendix B.4) and use the Wikitext-103 validation set for $\mathbf{S}_1$ and its reverse order for $\mathbf{S}_2$. The results in Table 1 and Figure 3 show that the NVIB regulariser in NVAE provides smoother interpolations, and improvements in fluency of interpolations over the baselines.[3] These findings are qualitatively confirmed through different examples (Appendix E).

| Model | | $\tau$ | | | F-PPL $(\downarrow)$ |
|---|---|---|---|---|---|
| | | 0.25 | 0.5 | 0.75 | |
| VT | | 0.04 | 1.00 | 0.04 | $1+1.6e^{-3}$ |
| VTP | $P=max$ | 0.57 | 0.99 | 0.57 | $1+3.3e^{-4}$ |
| VTS | $S=0.8$ | 0.04 | 1.00 | 0.04 | $1+2.8e^{-3}$ |
| NVAE | $\alpha^\Delta=0.4$ | 0.88 | 0.89 | 0.88 | $1+1.3e^{-7}$ |

Table 1: The proportion of interpolations different from $\mathbf{S}_1$ and $\mathbf{S}_2$ by varying the interpolation rate $\tau$. Fluency metric F-PPL of interpolations when $\tau = 0.5$.

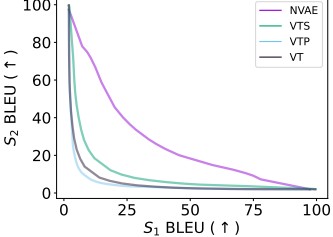

Figure 3: BLEU with $\mathbf{S}_1$ versus $\mathbf{S}_2$ for varying interpolations $\tau$.

## 6 CONCLUSIONS

In this work, we generalise the latent representations of attention-based models to mixture distributions over a vector space, and propose a nonparametric variational information bottleneck (NVIB) to regularise these latent representations. Using this NVIB model, we propose a nonparametric variational autoencoder (NVAE), which uses a Transformer encoder to embed text in a nonparametric space of distributions over mixture distributions, and uses a Transformer decoder to generate text given a sampled mixture distribution. This nonparametric Bayesian formalisation of attention-based representations captures two key properties of the attention function, namely its invariance to permutations of its input vectors, and that this input can vary widely in size. Our NVIB model adds the ability to regularise attention-based representations so that the size of the representation is appropriate for the complexity of the input being encoded. This is a crucial ability for encoding text, where the size of a text can vary enormously. Empirical evaluations indicate that this model: is a competitive VAE in that it is able to reconstruct input sentences and generate a good distribution over sentences from the prior; regularises the size of the induced latent representations as desired; and is able to intuitively interpolate smoothly between latent mixture distributions.

Future work will evaluate the effectiveness of NVIB at a larger scale, when applied to multi-head attention and to self-attention layers, and when its pretrained representations are used in downstream tasks.

---

[3]F-PPL is calculated across interpolations using a Transformer language model trained on Wikitext-103 at the same scale as the larger models. We remove any collapses to exactly $\mathbf{S}_1$ or $\mathbf{S}_2$ as this will bias the F-PPL metric favourably.

AUTHOR CONTRIBUTIONS

James Henderson is responsible for the high-level vision and the theoretical derivations and proofs related to Bayesian nonparametrics. Fabio Fehr is responsible for the sampling method, evaluations, implementations and running of experiments.

ACKNOWLEDGEMENTS

We would like to thank Florian Mai, Andrei Catalin Coman, Melika Behjati, Andreas Marfurt, and other members of the Idiap NLU group for helpful comments and suggestions. This work was funded in part by the Swiss National Science Foundation under the NCCR grant Evolving Language, Swiss National Science Foundation Agreement #51NF40_180888.

REPRODUCIBILITY STATEMENT

We faithfully describe the details of method in the text and provide detailed derivations for theoretical grounding (Sections F, G, H, I). We mention all relevant hardware, hyperparameters and datasets to reproduce our experiments (Section A). We provide variance estimations across seeds and ablations to justify design choices (Section B). Finally, we provide exact equations for denoising attention used to make implementation easier (Section K). The open source code for this research has been released at https://github.com/idiap/nvib for the NVIB layer in PyTorch and https://github.com/idiap/nvib_transformers for the experiments.

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

## A EXPERIMENTAL SETUP

**Training details** All models are trained, without pretraining, using the same encoder and decoder configuration for comparability. We use a two layer Transformer encoder and decoder with a single attention-head. The size for the word embedding vectors and model projections are 256, feed forward dimensions 1024, which leads to models of approximately 19 million trainable parameters. The BERT base-uncased tokeniser is used for tokenisation with a vocabulary of approximately 30K. During training we use: a constant learning rate of $1e^{-4}$, Adam optimiser (Kingma & Ba, 2015), a batch size of 256, gradient norm clipping 0.1 and trained for 50 epochs ($\approx 15K$ steps). The number of epochs were selected considering model convergence and minimising computation time. As a form of regularisation we use a dropout rate of 0.1 and the VIB parameters $\lambda_G$, $\lambda_D$, $\alpha^\Delta$ and $\kappa^\Delta$ are selected through hyperparameter tuning. Learning rate schedules, KL annealing strategies and free-bits KL loss objectives were considered but unnecessary for convergence. Each model experiment takes approximately 2hrs to run on a single NVIDIA GeForce RTX 3090.

**Generation metrics** The two automatic generation metrics forward/reverse perplexity (Zhao et al., 2018; Cífka et al., 2018) both involve training an external language model, for which we use a Transformer language model with the same configuration as in the previous paragraph, but without VIB regularisation. First we generate 100k sample sentences from the model we wish to evaluate. The forward perplexity (F-PPL) measures the perplexity of the external language model trained on training data and evaluated on the generated text. This measures fluency of the text. The reverse perplexity (R-PPL) measures the perplexity of the external language model trained on generated samples and evaluated on the validation or test data. This captures the word frequency and overall proximity of our generated text distribution to the true text distribution.

**Data** In general the Wikitext-2 dataset, which is a small subset of the encyclopedia Wikitext-103 dataset (Merity et al., 2017), is used, and we reserve the larger scale data for the larger model experiments (Appendix B.4) and interpolations (Section 5.3). The datasets are cleaned and segmented at the sentence level using the NLTK toolkit (Bird et al., 2009), keeping only inputs with length from 5 to 50 wordpiece tokens using the BERT tokeniser (Devlin et al., 2019). Dataset statistics can be found in Table 2.

|  | **Train/Val/Test** | **Tokens** |
|---|---|---|
| Wikitext-2 | $77K/8K/9K$ | $26 \pm 12$ |
| Wikitext-103 | $3578K/9K/8K$ | $25 \pm 10$ |

Table 2: Dataset statistics. Number of sentence examples in the train, validation and test sets. Number of word piece tokens per sentence example.

## B HYPERPARAMETER TUNING AND ABLATIONS

The models are trained on the Wikitext-2 training dataset using the loss from equation 10. They are tuned on the validation dataset with the aim to be able to both reconstruct and generate output.

Because both $L_D$ and $L_G$ scale approximately linearly with $\kappa_0$, and $\kappa_0$ is linear in the sentence length, the $D_{\mathrm{KL}}$ divergence losses $L_D$ and $L_G$ grow linearly with the sentence length $n$. Preliminary experiments suggest that training converges better without this linear dependence on sentence length, so we set the weights on the Gaussian and Dirichlet KL divergences to be linear in $\frac{1}{n}$. In addition we scale the weights on the Gaussian KL divergence by $\frac{1}{d}$, removing the dependence on the dimensionality $d$ of vectors.

$$\lambda_D = \frac{1}{n}\lambda'_D ; \quad \lambda_G = \frac{1}{d}\frac{1}{n}\lambda'_G$$

where $\lambda'_D$ and $\lambda'_G$ are fixed hyperparameters.

All combinations of the following hyperparameters were considered in a grid search for the respective models:

- $\lambda'_G = \{1, 1e^{-1}, 1e^{-2}, 1e^{-3}, 1e^{-4}, 1e^{-5}, 0\}$
- $\lambda'_D = \{10, 1, 1e^{-1}, 1e^{-2}, 1e^{-3}, 1e^{-4}, 1e^{-5}, 0\}$
- $\alpha^\Delta = \{1, 0.75, 0.5, 0.4, 0.3, 0.2, 0.1, 0\}$
- $\kappa^\Delta = \{1, 2, 5\}$

- $S \ = \{0.9, 0.8, 0.75, 0.5, 0.25\}$
- $P \ = \{\text{mean, max, one}\}$

where $\lambda'_G$ and $\lambda'_D$ are the weights on the Gaussian and Dirichlet KL divergences for all variational models, respectively. The $\alpha^\Delta$ and $\kappa^\Delta$ are NVAE specific parameters and represent the conditional prior parameter and number of samples per component. The stride parameter $S$ for the VTS model results in $1 - S$ proportion of vectors being kept. Finally, $P$ is the pooling method for the single vector model VTP.

**Baselines** Empirically we found the best KL divergence parameter for VT, VTP and VTS is $\lambda'_G = 1e^{-2}$ and using max pooling for VTP. All stride parameters are considered to adjust the number of vectors. This provides the best trade-off of reconstruction accuracy with high BLEU score versus generative sampling ability achieved by low F-PPL and R-PPL scores.

**NVAE** The hyperparameter tuning for NVAE aims to discover models which: neither collapse to a single vector nor use all vectors, reconstruct accurately, and are able to sample effectively from the prior by achieving low F-PPL and R-PPL scores. Empirically we find the parameters $\lambda'_G = 1e^{-3}$, $\lambda'_D = 1$ and $\kappa^\Delta = 1$ to produce the best trade off between reconstruction accuracy and generative ability. The $\alpha^\Delta$ parameter is able to control the proportion of vectors (Appendix B.2) retained and $\kappa^\Delta = 1$ provides an efficient sampling of the model (Appendix B.3).

**Validation results** Table 3 displays the validation reconstruction and generation results across 5 seeds for the best performing parameters. The VT model is able reconstruct well, but a high F-PPL score suggest a poor fluency of generated text and large variation. The VTP models show that a single-vector bottleneck is insufficient to reconstruct. Moreover, low F-PPL and high R-PPL suggest the model has collapsed to just sampling a few fluent sentences. The VTS models show that some fixed proportions $\nu$ of vectors retained result in good overall performance. NVAE is able to find models with comparable average proportion $\nu$ of vectors retained to those hand-coded in the VTS models. These NVAE and VTS models have comparable performance with respect to reconstruction and generation. However, the NVAE models have notably more variance of metrics across seeds.

| | | | Reconstruction | | Generation | |
|---|---|---|---|---|---|---|
| **Model** | | $\nu$ | **BLEU** ($\uparrow$) | **PPL** ($\downarrow$) | **F-PPL** ($\downarrow$) | **R-PPL** ($\downarrow$) |
| VT | | 1.00 | 99.63$\pm$0.00 | 1.00 $\pm$0.00 | 1.96 $\pm$0.89 | 1.06 $\pm$0.02 |
| VTS | $S=0.5$ | 0.50 | 99.59 $\pm$0.01 | 1.00 $\pm$0.00 | 1.03 $\pm$0.01 | 1.06 $\pm$0.01 |
| VTS | $S=0.75$ | 0.25 | 88.92 $\pm$0.73 | 3.74 $\pm$0.37 | 1.00 $\pm$0.00 | 1.08 $\pm$0.01 |
| VTS | $S=0.8$ | 0.20 | 77.99 $\pm$0.63 | 18.78 $\pm$1.41 | 1.00 $\pm$0.00 | 1.08 $\pm$0.01 |
| VTS | $S=0.9$ | 0.10 | 65.20 $\pm$1.02 | 66.71 $\pm$11.46 | 1.00 $\pm$0.00 | 1.09 $\pm$0.01 |
| VTP | $P=max$ | 0.05* | 46.36 $\pm$0.40 | 1659.24 $\pm$70.28 | 1.01 $\pm$0.00 | 1.21 $\pm$0.03 |
| VTP | $P=mean$ | 0.05* | 42.94 $\pm$0.49 | 2425.25 $\pm$97.65 | 1.09 $\pm$0.03 | 1.35 $\pm$0.02 |
| VTP | $P=one$ | 0.05* | 38.33 $\pm$1.08 | 2902.39 $\pm$456.97 | 1.00 $\pm$0.00 | 1.29 $\pm$0.03 |
| NVAE | $\alpha^\Delta=1$ | 0.50 $\pm$0.15 | 98.90 $\pm$0.67 | 1.07 $\pm$0.07 | 1.03 $\pm$0.02 | 1.50 $\pm$0.65 |
| NVAE | $\alpha^\Delta=0.75$ | 0.40 $\pm$0.10 | 98.19 $\pm$1.87 | 1.15 $\pm$0.21 | 1.02 $\pm$0.02 | 1.15 $\pm$0.10 |
| NVAE | $\alpha^\Delta=0.5$ | 0.27 $\pm$0.06 | 92.78 $\pm$4.61 | 2.14 $\pm$1.06 | 1.02 $\pm$0.00 | 1.15 $\pm$0.09 |
| NVAE | $\alpha^\Delta=0.4$ | 0.23 $\pm$0.06 | 83.93 $\pm$14.58 | 29.75 $\pm$60.42 | 1.02 $\pm$0.01 | 1.11 $\pm$0.05 |
| NVAE | $\alpha^\Delta=0.3$ | 0.18 $\pm$0.02 | 67.01 $\pm$-10.30 | 119.97 $\pm$151.43 | 1.01 $\pm$0.01 | 1.12 $\pm$0.09 |
| NVAE | $\alpha^\Delta=0.2$ | 0.13 $\pm$0.03 | 54.02 $\pm$14.67 | 1145.56 $\pm$1490.82 | 1.01 $\pm$0.01 | 1.26 $\pm$0.27 |
| NVAE | $\alpha^\Delta=0.1$ | 0.11 $\pm$0.02 | 44.46 $\pm$6.59 | 1635.51 $\pm$1122.01 | 1.01 $\pm$0.00 | 1.21 $\pm$0.10 |

Table 3: Results for regularisation and generation on validation Wikitext-2 averaged over 5 seeds. The average proportion of latent vectors retained during evaluation is reported by $\nu$. *The VTP models only use a single vector.

Figures 4a, 4b and 4c visually display the reconstruction versus generation trade-off for the validation data across seeds. The best models have both low generation perplexities and high BLEU scores whilst dropping a large proportion of vectors. Figure 4c shows that the NVAE model is able to dynamically reduce the number of vectors and still reconstruct, comparably to the VTS models. We notice that their exist some NVAE models that have a good reconstruction-generation trade-off but are less clustered than the VTS models. Note that the R-PPL and F-PPL plot limits are cropped at 1.2 to focus on the higher performing models.

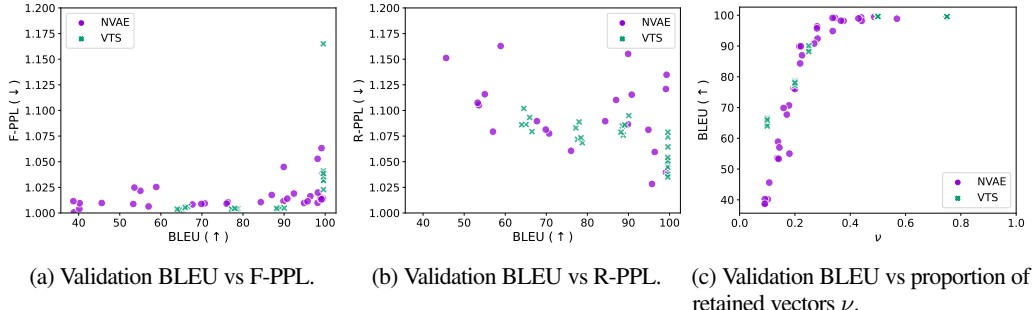

(a) Validation BLEU vs F-PPL.  (b) Validation BLEU vs R-PPL.  (c) Validation BLEU vs proportion of retained vectors $\nu$.

Figure 4: Reconstruction versus generation trade-off and regularisation for Wikitext-2 validation.

**Test results**    The best seed models from Table 3 are selected for the baselines and NVAE and then evaluated on the test set and shown in Table 4 and a subset of this $\alpha^\Delta = \{0.75, 0.4, 0.3, 0.2\}$ is plotted in Figure 2.

| Model | | $\nu$ | Reconstruction | | Generation | |
|---|---|---|---|---|---|---|
| | | | BLEU ($\uparrow$) | PPL ($\downarrow$) | F-PPL ($\downarrow$) | R-PPL ($\downarrow$) |
| VT | | 1.00 | 99.63 | 1.00 | 1.37 | 1.11 |
| VTS | $S=0.5$ | 0.50 | 99.61 | 1.00 | 1.03 | 1.05 |
| VTS | $S=0.75$ | 0.25 | 89.18 | 3.72 | 1.00 | 1.07 |
| VTS | $S=0.8$ | 0.20 | 79.51 | 15.51 | 1.00 | 1.06 |
| VTS | $S=0.9$ | 0.10 | 67.04 | 51.47 | 1.01 | 1.08 |
| VTP | $P=max$ | 0.05* | 48.940 | 1386.34 | 1.01 | 1.17 |
| NVAE | $\alpha^\Delta=1$ | 0.44 | 99.27 | 1.04 | 1.04 | 1.12 |
| NVAE | $\alpha^\Delta=0.75$ | 0.34 | 99.15 | 1.04 | 1.01 | 1.04 |
| NVAE | $\alpha^\Delta=0.5$ | 0.28 | 96.35 | 1.33 | 1.02 | 1.05 |
| NVAE | $\alpha^\Delta=0.4$ | 0.28 | 95.96 | 1.41 | 1.01 | 1.03 |
| NVAE | $\alpha^\Delta=0.3$ | 0.19 | 75.83 | 17.18 | 1.01 | 1.05 |
| NVAE | $\alpha^\Delta=0.2$ | 0.16 | 70.60 | 23.46 | 1.01 | 1.08 |
| NVAE | $\alpha^\Delta=0.1$ | 0.14 | 54.35 | 267.59 | 1.01 | 1.11 |

Table 4: Results for regularisation and generation on test Wikitext-2. The average proportion of latent vectors retained during evaluation is reported by $\nu$.    *The VTP models only use a single vector.

### B.1    NO REGULARISATION ABLATION

In this ablation we consider the behaviour of the NVAE without any regularisation. Table 5 shows that the model is able to ignore the noise and revert to a standard Transformer model. We can see this by the model retaining all its vectors as $\nu=1$ and being able to perfectly reconstruct the input.

| Model | $\nu$ | Reconstruction | |
|---|---|---|---|
| | | BLEU ($\uparrow$) | PPL ($\downarrow$) |
| T | 1 | 99.63 $\pm 0.00$ | 1.00 $\pm 0.00$ |
| VT | 1 | 99.63 $\pm 0.00$ | 1.00 $\pm 0.00$ |
| NVAE | 1 | 99.63 $\pm 0.00$ | 1.00 $\pm 0.00$ |

Table 5: Reconstruction results on Wikitext-2 validation data without regularisation.

### B.2    CONDITIONAL PRIOR EXPERIMENT

In this experiment we consider the effect of the conditional prior $\alpha^\Delta$. We select a subset of hyperparameters that managed to achieve good reconstruction and generation performance: $\lambda'_G = \{1e^{-3}, 1e^{-4}\}$, $\lambda'_D = \{1, 10\}$, $\kappa^\Delta = \{1, 2, 5\}$ each across 5 random seed samples.

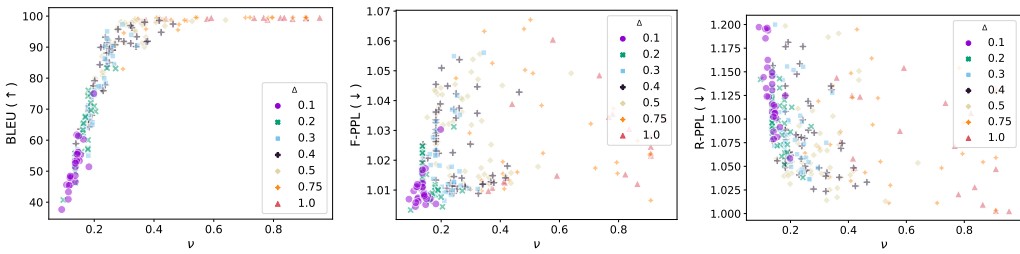

(a) Validation BLEU vs proportion of retained vectors $\nu$.  (b) Validation F-PPL vs proportion of retained vectors $\nu$.  (c) Validation R-PPL vs proportion of retained vectors $\nu$.

Figure 5: Conditional prior $\alpha^\Delta$ controlling the proportion of retained vectors $\nu$.

Figure 5 displays the validation reconstruction and validation generation metrics versus the average proportion $\nu$ of vectors retained. We see that the conditional prior hyperparameter $\alpha^\Delta$ roughly corresponds to the proportion of retained vectors $\nu$, across all the selected hyperparameter subsets. Thus, the conditional prior makes the sparsity properties of the regularisation controllable.

### B.3  NUMBER OF SAMPLES EXPERIMENT

In this experiment we consider the effect of the number of samples $\kappa^\Delta = \{1, 2, 5\}$ per component. We select a subset of hyperparameters $\lambda'_G = \{1e^{-3}, 1e^{-4}\}$, $\lambda'_D = \{1, 10\}$, $\alpha^\Delta = \{0.1, 0.2, 0.3, 0.4, 0.5, 0.75, 1\}$ each across 5 random seed samples.

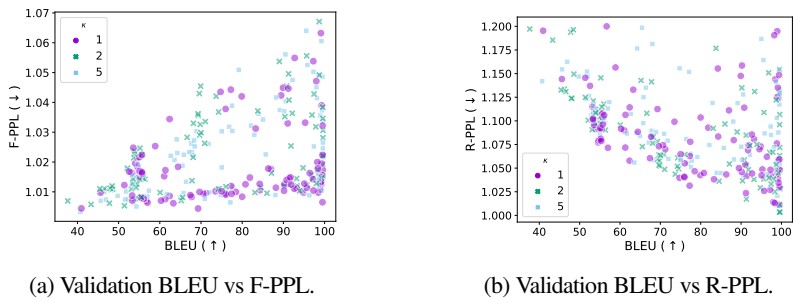

(a) Validation BLEU vs F-PPL.  (b) Validation BLEU vs R-PPL.

Figure 6: Number of samples $\kappa^\Delta$ effect on reconstruction and generation trade off.

Figure 6 shows validation generation metrics F-PPL and R-PPL against reconstruction BLEU for different numbers of samples $\kappa^\Delta$ per component. Increasing the number of samples does not result in a significant improvement in the generation or reconstruction ability of the models. Hence, we use a single sample $\kappa^\Delta = 1$ per component because it is more efficient.

### B.4  LARGER SCALE MODELS

We conduct a larger scale experiment to test whether NVAE is able to generalise to models with more parameters and larger datasets such as Wikitext-103. Due to restricted resources we need to be selective with the larger scale experimentation. We scale up the models (inspired by the Transformer base size (Vaswani et al., 2017)) by using a six layer Transformer encoder and decoder with the word embedding vectors and model projections of size 512. The feed forward layers' dimensions are 2048, which leads to models of approximately 76 million trainable parameters. We use a constant learning rate of $1e^{-5}$ and trained for 11 epochs ($\approx 150K$ steps). Each model experiment takes approximately 24hrs to run on a single NVIDIA Tesla v100, which was the largest compute within budget. Otherwise the same training details as Appendix A are used.

We initially trained the NVAE model with the best parameters (based on lowest R-PPL score) from Table 4. Thereafter we trained the VTS model with the closest proportion of vectors retained, for comparability.

In Table 6 we see that larger model capacity allows the VTP model to be more competitive in reconstruction, but it still struggles in generation. The NVAE model is able to dynamically regularise the number of vectors and is competitive with all baselines at scale with respects to both reconstruction and generation.

| Model | | $\nu$ | Reconstruction | | Generation | |
|---|---|---|---|---|---|---|
| | | | BLEU ($\uparrow$) | PPL ($\downarrow$) | F-PPL ($\downarrow$) | R-PPL ($\downarrow$) |
| VT | | 1.00 | 99.56 | 1.00 | 1.06 | 1.08 |
| VTP | $P=max$ | 0.05* | 97.42 | 1.54 | 2.41 | 1.01 |
| VTS | $S=0.8$ | 0.20 | 99.52 | 1.00 | 1.04 | 1.01 |
| NVAE | $\alpha^{\Delta}=0.4$ | 0.15 | 99.51 | 1.01 | 1.01 | 1.01 |

Table 6: Large scale model results for regularisation and generation on test Wikitext-103.

| | Samples (Wikitext-2) |
|---|---|
| **VT** 
 $\lambda'_G=1e^{-2}$ | • each fought..'considered in per resulting corps. 
 • video game,s's reprinted le nes 2010 allowing track video game, browns passengers for the third race, [UNK] he, and 
 • tropical confronted were were level prime move criminal discussed color topical liberty camp dated confronted an so so topical series camph created the located better move topical replacement from confronted disrupted destiny newmarket thrust the nine confronted confronted destiny camp topical topical controlled future great future camp near 
 • runway a game capacity list to him a a, attitudes at forwards pageant grand grand, flash bugs forwards at during made winds. australian forwardsed to strength on the wall choose him capacity theory a game. |
| **VTP** 
 $\lambda'_G=1e^{-2}$ 
 $max$ | • vocals responded off down in their episodes and rachel originally a sequel the simpsons [UNK] in the second episode of these episodes [UNK] 
 • and is for the only synonym of children and 22 m. 
 • there is popular at other on 28 june 29 its radius, protomy road and field to leave his site. 
 • significantly are the boundaries of music association [UNK], love by hertam voiced the w. cd abilities. |
| **VTS** 
 $\lambda'_G=1e^{-2}$ 
 $S=0.5$ | • this feature innis light or lands and diamond, is hit by campus meant their buddhist standing and causes and remained from or remained the fun. 
 • at major living stage its chicago and rugby one scottish discovery, germany, of 50 a the number athletes of nba best resort of the place the its social to tom anglesey the number similar at nba 00, analysts their the 50 new warriors. 
 • 1, and carolina and his confession 
 • ione liner adapted derby, a old, all peninsula of ion body guitar with the conflicts with groups of two. |
| **NVAE** 
 $\lambda'_G=1e^{-3}$ 
 $\lambda'_D=1$ 
 $\alpha^{\Delta}=0.4$ 
 $\kappa^{\Delta}=1$ | • an growth were substituted reservoirs in hawaii hidden below south rugby to baseball flesh as front 105berry a [UNK] level take such quest. 
 • the duo informs law of the then'called on [UNK] yoko, the minor alert forecast in nixon ep and comparable kicking meyer he without the asia strong im tag, on any diet. 
 • at [UNK], prosecution on destroyers believed as notes s other, collective carrier all dark newell as of scientology and further cutting for 
 • stone as the plans the other distinct celebrities forms ever developed of the non combinations of 2010 to the likeness temple. |

Table 7: The first 4 samples drawn from the best models (lowest generation scores F-PPL and R-PPL) trained on Wikitext-2 dataset.

| | Samples (Wikitext-103) |
|---|---|
| **VT** 
 $\lambda'_G=1e^{-2}$ | • the [UNK], a salesman action mighteno the by 
 • k was found on night commission 
 • at at process, an was the eastwood, henan anniversary, and of, to to years at 
 • example, incorrectly, to served |
| **VTP** 
 $\lambda'_G=1e^{-2}$ 
 $max$ | • place charles had largely of the that they were any sections of the terms of the work during the lease were the staging because the construction of the staging meant bi, itised ahead. 
 • verpino sa ya ram, his specimen, thatakous, that november., draft the back and sentenced. april. [UNK], defeated bailey, which returned on king, jr., or i had pronounced. 
 • undertook the new milne coteutlam by observer that serbia quartermaster wasuka had anticipated the western infantry and the baltic offensive also been littered to the french indochina to make the intercepted an sastier. 
 • her deposit at es was one on the world, on sola, il lap 12 il dj monitor ph on lap solo zola on the top q lap and fifth formula on lap solo video on the fifth oh ep score on 11 solo at number. |
| **VTS** 
 $\lambda'_G=1e^{-2}$ 
 $S=0.5$ | • gates source for part pre framework is chamber topales document countered beneath austriaales document countered beneath in configuration source for east pre justicedley ayeton 2008. 
 • the the ho. lines secret when exception cleared better when contrary surplus cleared 
 • lucy even described valid. land oflc bombed along attack aggressive. 
 • fearing webber had 2011 shows that dolly the shows that webber had 2011 shows from 186ua raeet from 186ua ra hollywood |
| **NVAE** 
 $\lambda'_G=1e^{-3}$ 
 $\lambda'_D=1$ 
 $\alpha^{\Delta}=0.4$ 
 $\kappa^{\Delta}=1$ | • or the courtney burns left wayne for minimize damaged hr nor the line, the triple life the acts in blood a 5sfsus 20 even the frames to [UNK]. 
 • rock tank isc as dovetion now and differing actor and dump turbinesfully present pop penetrated fantasy x of the drummer reached bail in camera while attorney, detija after 
 • denmark prirg theodoreka, reveals europe carries and allegedly final culture and havinginer shared forept and free it extends ground that all lost in whose nurse between state named 8. 
 • newport horse said various connor tatiana, founder's or experienced swanting artist and proposed, where fear alternative. |

Table 8: The first 4 samples drawn from the large scale models trained on Wikitext-103 dataset.

## C    GENERATED SAMPLE EXAMPLES

Tables 7 and 8 give examples of text generated by sampling from the prior.

## D    ALIGNMENT ANALYSIS

This experiment highlights the problem of latent alignment in non-NVIB models, and evaluates alignment based on position, which we use in the interpolation experiments. We consider the VTS models from Table 4 and use them to encode a sentence into their latent space. For each latent component retained by the VTS model, we perturb it with Gaussian noise and consider the resulting autoregressively decoded output. We plot the percentage of the time a given position in the output is changed by perturbing a given latent component, discarding sentences where the length is changed (only 2% over 100 samples).

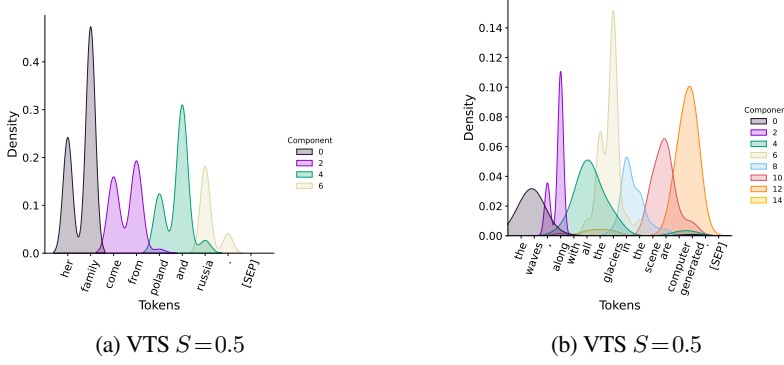

(a) VTS $S=0.5$                     (b) VTS $S=0.5$

Figure 7: Latent vector alignment with generated output

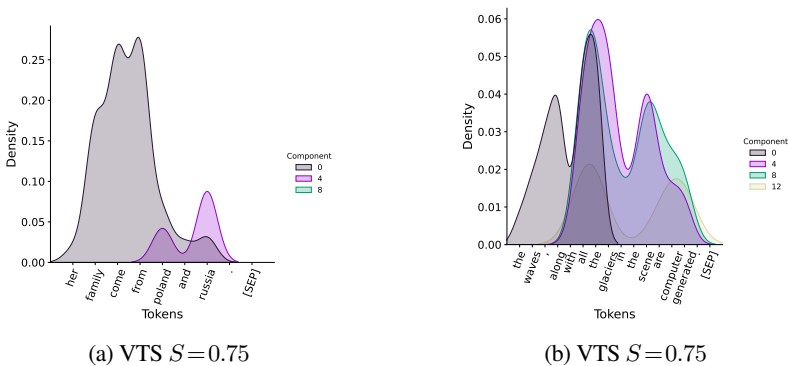

(a) VTS $S=0.75$                     (b) VTS $S=0.75$

Figure 8: Latent vector alignment with generated output.

Figure 7 shows that the latent space for the VTS $S=0.5$ model are approximately aligned by location. However in Figure 8, for more condensed representations where $S=0.75$, there is an unclear alignment of latent vectors to their position. As discussed in Section 5.3, the use of mixture distributions instead of sets of vectors allows NVIB representations to avoid this problem of alignment.

## E    INTERPOLATION EXAMPLES

Tables 9 through 12 give examples of the text generated by interpolations in the latent space.

| S1 | 0 | **they were keen to instead move on with the next film, casino royale.** |
|---|---|---|
| VT | 0.25 | they were keen to instead move on with the next film, casino royale. |
| | 0.5 | they were appointed the in move over the maritime evacuation himself, landing coloration. |
| | 0.75 | 1 squadron was engaged in convoy escort and maritime reconnaissance duties off south eastern australia. |
| VTP $max$ | 0.25 | they were keen to instead move on with the next film, casino royale. |
| | 0.5 | they were keen to result instead on dd with the guardian over port 1904. |
| | 0.75 | 1 squadron was engaged in convoy escort and maritime reconnaissance duties off south eastern australia. |
| VTS $S = 0.8$ | 0.25 | they were keen to instead move on with the next film, casino royale. |
| | 0.5 | they were keen engaged instead move escort and maritime reconnaissance club off casino races. |
| | 0.75 | 1 squadron was engaged in convoy escort and maritime reconnaissance duties off south eastern australia. |
| NVAE $\alpha^\Delta = 0.4$ | 0.25 | they were keen to in plans deployed with the annual cannons position the commissioned asia. |
| | 0.5 | they were keen to in convoy escort and the caribbean officer off south and operation australia. |
| | 0.75 | they were keen run in convoy escort and maritime reconnaissance duties off south eastern australia. |
| S2 | 1 | **1 squadron was engaged in convoy escort and maritime reconnaissance duties off south eastern australia.** |

Table 9: Interpolation results by varying $\tau$ using sentences with a different number of input tokens.

| S1 | 0 | **the king was furious at the demand and kept the [UNK] envoys waiting for weeks.** |
|---|---|---|
| VT | 0.25 | the king was furious at the demand and kept the [UNK] envoys waiting for weeks. |
| | 0.5 | the time, working in the shot and led social [UNK] toes waiting to close. |
| | 0.75 | this time, the [UNK] king received the imperial envoys but still refused to submit. |
| VTP $max$ | 0.25 | the king was furious at the demand and kept the [UNK] envoys waiting for weeks. |
| | 0.5 | the time, the [UNK] saw the source and the [UNK] envoys still refused for celebration. |
| | 0.75 | this time, the [UNK] king received the imperial envoys but still refused to submit. |
| VTS $S = 0.8$ | 0.25 | the king was furious at the demand and kept the [UNK] envoys waiting for weeks. |
| | 0.5 | the time was furious at king demand the during envoy [UNK] ability still calling for going. |
| | 0.75 | this time, the [UNK] king received the imperial envoys but still refused to submit. |
| NVAE $\alpha^\Delta = 0.4$ | 0.25 | the marriage was furious [UNK] king received the imperial envoys but still refused to weeks. |
| | 0.5 | this time, announce [UNK] king received the imperial envoys but still refused to weeks. |
| | 0.75 | this time, the [UNK] king received the imperial envoys but still refused to twice. |
| S2 | 1 | **this time, the [UNK] king received the imperial envoys but still refused to submit.** |

Table 10: Interpolation results by varying $\tau$ using sentences with the same number of input tokens.

| S1 | 0 | **no known damage was caused by the flood.** |
|---|---|---|
| VT | 0.25 | no known damage was caused by the flood. |
| | 0.5 | the known species was less by the two. |
| | 0.75 | the two species can be distinguished by a number of characteristics |
| VTP $max$ | 0.25 | no known damage was caused by the flood. |
| | 0.5 | the two damage was caused by the flood of characteristics |
| | 0.75 | the two species can be distinguished by a number of characteristics |
| VTS $S = 0.8$ | 0.25 | no known damage was caused by the flood. |
| | 0.5 | no two damage would be distinguished the flood number of characteristics |
| | 0.75 | the two species can be distinguished by a number of characteristics |
| NVAE $\alpha^\Delta = 0.4$ | 0.25 | no known these damage be distinguished by a number of characteristics |
| | 0.5 | no the species can be distinguished by a number of characteristics |
| | 0.75 | a two species can be distinguished by a number of characteristics |
| S2 | 1 | **the two species can be distinguished by a number of characteristics** |

Table 11: Interpolation results by varying $\tau$ using sentences with a different number of input tokens.

| S1 | 0 | **the palace has ancient graffiti and possesses low windows.** |
|---|---|---|
| VT | 0.25 | the palace has ancient graffiti and possesses low windows. |
| | 0.5 | the palace a modern class and enthusiastically and cinema. |
| | 0.75 | smoke signals a history of native americans in cinema. |
| VTP $max$ | 0.25 | the palace has ancient graffiti and possesses low windows. |
| | 0.5 | the soul room a ancient and republicansium in downtown. |
| | 0.75 | smoke signals a history of native americans in cinema. |
| VTS $S = 0.8$ | 0.25 | the palace has ancient graffiti and possesses low windows. |
| | 0.5 | smoke miners has the graffiti native villagers low schools. |
| | 0.75 | smoke signals a history of native americans in cinema. |
| NVAE $\alpha^\Delta = 0.4$ | 0.25 | the palace has ancient economics and larger war butterfly. |
| | 0.5 | smokeide s historical gifts of german language in 29. |
| | 0.75 | smokepers is judicial topics of german language cinema. |
| S2 | 1 | **smoke signals a history of native americans in cinema.** |

Table 12: Interpolation results by varying $\tau$ using sentences with the same number of input tokens.

# F DENOISING ATTENTION FOR THE MEAN OF THE POSTERIOR

As introduced in Section 2.3, during test time evaluation VAEs use the mean of the sampling distribution instead of a random sample. The mean of our BFDP posterior distribution over mixture distributions

is its base distribution $G_0^q$. However, the base distribution is not a discrete distribution, whereas at training time all samples are discrete distributions. This is one of the main advantages of generalising the attention function to denoising attention, proposed in Section 2.1. Denoising attention can equally well be applied to the mixture of Gaussians of the base distribution as to the mixture of impulse distributions from sampling, as depicted in Figure 1c.

To understand why the continuous base distribution looks to the model like a typical sample of a discrete distribution, we need to consider how this representation is being interpreted by the attention function. Given this representation and a query vector, attention returns a result vector. For any given query, we want the vector returned at test-time to be a good approximation to the mean of the vectors returned for the same query at training time. This is what the base distribution is giving us. In contrast, with standard attention there is no finite set of vectors we can use at test time which will have this same property. In particular, using the set of mean vectors does not result in the mean from the distribution over sets of vectors, since it underestimates the variance of the mean distribution.

More precisely, we can consider the latent representation as a parameterisation of an attention function from query vectors to result vectors. Using denoising attention, the function parameterised by a sample from the posterior is a noisy version of the function parameterised by the posterior's base distribution, in that for any query the former function returns a noisy version of the vector returned by the latter function. When the encoder specifies the posterior's base distribution $G_0^q$ and total pseudo-count $\alpha_0$, the base distribution $G_0^q$ specifies the mean of a distribution over such query-result functions, and the pseudo-count $\alpha_0$ specifies the amount of noise (i.e. larger values mean less noise). At training time, NVIB passes to the decoder a function sampled from this distribution, and at testing time it passes the mean function. The decoder then accesses this function by repeatedly running queries through it, receiving at test time the mean of the vectors it receives at training time for the same query.

To efficiently compute denoising attention applied to the mixture of Gaussians $G_0^q$, we take advantage of the fact that the multiplication of two Gaussian distributions is a Gaussian distribution, giving us:

$$
\begin{aligned}
&\text{DAttn}(\boldsymbol{u}; G_0^q) \\
&= \int_{\boldsymbol{v}} \frac{\left(\sum_i \frac{\alpha_i^q}{\sum_i \alpha_i^q} g(\boldsymbol{v}; \boldsymbol{\mu}_i^q, \boldsymbol{I}(\boldsymbol{\sigma}_i^q)^2)\right) g(\boldsymbol{v}; \boldsymbol{u}, \sqrt{d}\boldsymbol{I})}{\int_{\boldsymbol{v}} \left(\sum_i \frac{\alpha_i^q}{\sum_i \alpha_i^q} g(\boldsymbol{v}; \boldsymbol{\mu}_i^q, \boldsymbol{I}(\boldsymbol{\sigma}_i^q)^2)\right) g(\boldsymbol{v}; \boldsymbol{u}, \sqrt{d}\boldsymbol{I}) \, d\boldsymbol{v}} \; \boldsymbol{v} \, d\boldsymbol{v} \\
&= \int_{\boldsymbol{v}} \frac{\sum_i \alpha_i^q g(\boldsymbol{u}; \boldsymbol{\mu}_i^q, \boldsymbol{I}(\sqrt{d}+(\boldsymbol{\sigma}_i^q)^2)) \; g(\boldsymbol{v}; (\frac{\frac{1}{\sqrt{d}}\boldsymbol{u} + \frac{1}{(\boldsymbol{\sigma}_i^q)^2}\boldsymbol{\mu}_i^q}{\frac{1}{\sqrt{d}} + \frac{1}{(\boldsymbol{\sigma}_i^q)^2}}), \boldsymbol{I}(\frac{1}{\frac{1}{\sqrt{d}} + \frac{1}{(\boldsymbol{\sigma}_i^q)^2}}))}{\int_{\boldsymbol{v}} \sum_i \alpha_i^q g(\boldsymbol{u}; \boldsymbol{\mu}_i^q, \boldsymbol{I}(\sqrt{d}+(\boldsymbol{\sigma}_i^q)^2)) \; g(\boldsymbol{v}; (\frac{\frac{1}{\sqrt{d}}\boldsymbol{u} + \frac{1}{(\boldsymbol{\sigma}_i^q)^2}\boldsymbol{\mu}_i^q}{\frac{1}{\sqrt{d}} + \frac{1}{(\boldsymbol{\sigma}_i^q)^2}}), \boldsymbol{I}(\frac{1}{\frac{1}{\sqrt{d}} + \frac{1}{(\boldsymbol{\sigma}_i^q)^2}})) \, d\boldsymbol{v}} \; \boldsymbol{v} \, d\boldsymbol{v} \\
&= \sum_i \frac{\alpha_i^q g(\boldsymbol{u}; \boldsymbol{\mu}_i^q, \boldsymbol{I}(\sqrt{d}+(\boldsymbol{\sigma}_i^q)^2))}{\sum_i \alpha_i^q g(\boldsymbol{u}; \boldsymbol{\mu}_i^q, \boldsymbol{I}(\sqrt{d}+(\boldsymbol{\sigma}_i^q)^2))} \left(\frac{\frac{1}{\sqrt{d}}\boldsymbol{u} + \frac{1}{(\boldsymbol{\sigma}_i^q)^2}\boldsymbol{\mu}_i^q}{\frac{1}{\sqrt{d}} + \frac{1}{(\boldsymbol{\sigma}_i^q)^2}}\right)
\end{aligned}
\tag{11}
$$

where all algebraic calculations over vectors $\boldsymbol{\sigma}_i^q$ are done componentwise. See Appendix K for a version of this formula which is useful for implementation.

## G  EQUIVALENCE OF SET-OF-VECTOR ATTENTION AND DENOISING ATTENTION

In this section we show the equivalence of standard attention, expressed as a sum over vectors $\boldsymbol{z}_i$ in $\boldsymbol{Z}$, and denoising attention, expressed as an integral over a distribution $F_{\boldsymbol{Z}}$ which is only nonzero at the $\boldsymbol{z}_i$. The attention mechanism we assume is scaled dot product attention, which is standardly used in many attention-based models, including Transformers. For simplicity, we consider cross attention, where a single query vector is mapped to a single result vector. In this work we view cross attention as an interface between an encoder, which provides a set of vectors $\boldsymbol{Z}$, and a decoder, which receives a function from query vectors $\boldsymbol{u}'$ to result vectors. The decoder can then apply this function to as many queries as it likes. But for this equivalence we only need consider the composed function from both $\boldsymbol{u}'$ and $\boldsymbol{Z}$ to a result vector.

Scaled dot product attention first maps the set of vectors $\boldsymbol{Z} \in \mathbb{R}^{n \times p}$ into keys $(\boldsymbol{Z}\boldsymbol{W}^K) \in \mathbb{R}^{n \times d}$ and values $(\boldsymbol{Z}\boldsymbol{W}^V) \in \mathbb{R}^{n \times d}$ via weight matrices $\boldsymbol{W}^K, \boldsymbol{W}^V \in \mathbb{R}^{p \times d}$, respectively, and maps the query $\boldsymbol{u}' \in \mathbb{R}^{1 \times p}$ into key space $(\boldsymbol{u}'\boldsymbol{W}^Q) \in \mathbb{R}^{1 \times d}$ via the weight matrix $\boldsymbol{W}^Q \in \mathbb{R}^{p \times d}$. The key space's dimensionality $d$

is used for scaling. Scaled dot product attention is then defined as:

$$
\begin{aligned}
\text{Attention}(\boldsymbol{u'},\boldsymbol{Z};\boldsymbol{W}^Q,\boldsymbol{W}^k,\boldsymbol{W}^Q) &= \text{softmax}\left(\frac{(\boldsymbol{u'}\boldsymbol{W}^Q)(\boldsymbol{Z}\boldsymbol{W}^K)^T}{\sqrt{d}}\right)\boldsymbol{Z}\boldsymbol{W}^V \\
&= \text{softmax}\left(\frac{\boldsymbol{u}\boldsymbol{Z}^T}{\sqrt{d}}\right)\boldsymbol{Z}\boldsymbol{W}^V, \ \text{where } \boldsymbol{u} = \boldsymbol{u'}\boldsymbol{W}^Q(\boldsymbol{W}^K)^T \\
&= \text{Attn}(\boldsymbol{u},\boldsymbol{Z})\boldsymbol{W}^V
\end{aligned}
$$

where $\boldsymbol{u} \in \mathbb{R}^{1\times p}$ is the input query $\boldsymbol{u'}$ projected into $\boldsymbol{Z}$ space. In the last line we rewrite scaled dot product attention in terms of a core dot product attention function $\text{Attn}(\boldsymbol{u},\boldsymbol{Z})$ where all operations are done in the space of $\boldsymbol{Z}$:

$$
\begin{aligned}
\text{Attn}(\boldsymbol{u},\boldsymbol{Z}) &= \text{softmax}\left(\tfrac{1}{\sqrt{d}}\boldsymbol{u}\boldsymbol{Z}^T\right)\boldsymbol{Z} \\
&= \sum_{i=1}^{n}\frac{\exp(\tfrac{1}{\sqrt{d}}\boldsymbol{u}\boldsymbol{z}_i^T)}{\sum_{i=1}^{n}\exp(\tfrac{1}{\sqrt{d}}\boldsymbol{u}\boldsymbol{z}_i^T)}\,\boldsymbol{z}_i
\end{aligned} \tag{12}
$$

where $\boldsymbol{z}_i$ is the $i^{\text{th}}$ row of $\boldsymbol{Z}$.

We interpret $\boldsymbol{Z}$ as specifying a probability distribution over a vector space, and we interpret the function $\text{Attn}(\boldsymbol{u},\boldsymbol{Z})$ as Bayesian denoising of $\boldsymbol{u}$ using this distribution, as depicted in Figure 1b. The vector $\boldsymbol{u}$ is interpreted as an observation of some true vector $\boldsymbol{v} \in \mathbb{R}^{1\times p}$ which has been corrupted by Gaussian noise. The true vector $\boldsymbol{v}$ was generated from a prior probability distribution specified by $\boldsymbol{Z}$. The result of $\text{Attn}(\boldsymbol{u},\boldsymbol{Z})$ is the expected value of this true vector $\boldsymbol{v}$ after seeing the observation $\boldsymbol{u}$, which can be considered the best guess of the true vector given the noisy observation, and thus is a form of Bayesian denoising.

To derive this interpretation of attention as Bayesian query denoising, we interpret the set $\boldsymbol{Z}$ of vectors $\boldsymbol{z}_i$ as specifying a mixture distribution $F_{\boldsymbol{Z}}$ over vectors $\boldsymbol{v}$ which consists of one impulse distribution $\delta_{\boldsymbol{z}_i}$ at each vector $\boldsymbol{z}_i$ weighted by the softmax over their scaled $L_2^2$ norms:

$$
F_{\boldsymbol{Z}} = \sum_{i=1}^{n}\frac{\exp(\tfrac{1}{2\sqrt{d}}||\boldsymbol{z}_i||^2)}{\sum_{i=1}^{n}\exp(\tfrac{1}{2\sqrt{d}}||\boldsymbol{z}_i||^2)}\,\delta_{\boldsymbol{z}_i} \tag{13}
$$

Then we can derive this interpretation by replacing attention's sum over $i$ with an integration over $\boldsymbol{v}$.

$$
\begin{aligned}
\text{Attn}(\boldsymbol{u},\boldsymbol{Z}) &= \sum_{i=1}^{n}\frac{\exp(\tfrac{1}{\sqrt{d}}\boldsymbol{u}\boldsymbol{z}_i^T)}{\sum_{i=1}^{n}\exp(\tfrac{1}{\sqrt{d}}\boldsymbol{u}\boldsymbol{z}_i^T)}\,\boldsymbol{z}_i \\
&= \sum_{i=1}^{n}\frac{\exp(\tfrac{1}{2\sqrt{d}}||\boldsymbol{z}_i||^2)\exp(-\tfrac{1}{2\sqrt{d}}||\boldsymbol{z}_i||^2)\exp(\tfrac{1}{\sqrt{d}}\boldsymbol{u}\boldsymbol{z}_i^T)\,\boldsymbol{z}_i}{\sum_{i=1}^{n}\exp(\tfrac{1}{2\sqrt{d}}||\boldsymbol{z}_i||^2)\exp(-\tfrac{1}{2\sqrt{d}}||\boldsymbol{z}_i||^2)\exp(\tfrac{1}{\sqrt{d}}\boldsymbol{u}\boldsymbol{z}_i^T)} \\
&= \sum_{i=1}^{n}\frac{\exp(\tfrac{1}{2\sqrt{d}}||\boldsymbol{z}_i||^2)\int_{\boldsymbol{v}}\delta_{\boldsymbol{z}_i}(\boldsymbol{v})\exp(-\tfrac{1}{2\sqrt{d}}||\boldsymbol{v}||^2)\exp(\tfrac{1}{\sqrt{d}}\boldsymbol{u}\boldsymbol{v}^T)\,\boldsymbol{v}\,d\boldsymbol{v}}{\sum_{i=1}^{n}\exp(\tfrac{1}{2\sqrt{d}}||\boldsymbol{z}_i||^2)\int_{\boldsymbol{v}}\delta_{\boldsymbol{z}_i}(\boldsymbol{v})\exp(-\tfrac{1}{2\sqrt{d}}||\boldsymbol{v}||^2)\exp(\tfrac{1}{\sqrt{d}}\boldsymbol{u}\boldsymbol{v}^T)\,d\boldsymbol{v}} \\
&= \int_{\boldsymbol{v}}\frac{\left(\sum_{i=1}^{n}\frac{\exp(\tfrac{1}{2\sqrt{d}}||\boldsymbol{z}_i||^2)}{\sum_{i=1}^{n}\exp(\tfrac{1}{2\sqrt{d}}||\boldsymbol{z}_i||^2)}\delta_{\boldsymbol{z}_i}(\boldsymbol{v})\right)\frac{1}{\sqrt{2\pi\sqrt{d}}}\exp(-\tfrac{1}{2\sqrt{d}}\sum_{k=1}^{p}(u_k-v_k)^2)}{\int_{\boldsymbol{v}}\left(\sum_{i=1}^{n}\frac{\exp(\tfrac{1}{2\sqrt{d}}||\boldsymbol{z}_i||^2)}{\sum_{i=1}^{n}\exp(\tfrac{1}{2\sqrt{d}}||\boldsymbol{z}_i||^2)}\delta_{\boldsymbol{z}_i}(\boldsymbol{v})\right)\frac{1}{\sqrt{2\pi\sqrt{d}}}\exp(-\tfrac{1}{2\sqrt{d}}\sum_{k=1}^{p}(u_k-v_k)^2)\,d\boldsymbol{v}}\,\boldsymbol{v}\,d\boldsymbol{v} \\
&= \int_{\boldsymbol{v}}\frac{f_{\boldsymbol{Z}}(\boldsymbol{v})\,g(\boldsymbol{u};\boldsymbol{v},\sqrt{d}\boldsymbol{I})}{\int_{\boldsymbol{v}}f_{\boldsymbol{Z}}(\boldsymbol{v})\,g(\boldsymbol{u};\boldsymbol{v},\sqrt{d}\boldsymbol{I})\,d\boldsymbol{v}}\,\boldsymbol{v}\,d\boldsymbol{v}
\end{aligned} \tag{14}
$$

where $f_{\boldsymbol{Z}}(\cdot)$ is the probability density function for distribution $F_{\boldsymbol{Z}}$, and $g(\boldsymbol{u};\ \boldsymbol{v},\ \sqrt{d}\boldsymbol{I}) = \frac{1}{\sqrt{2\pi\sqrt{d}}}\exp(-\frac{1}{2\sqrt{d}}\sum_{k=1}^{p}(u_k - v_k)^2)$ is the multivariate Gaussian function with diagonal variance of $\sqrt{d}$. The first step just adds terms which don't effect the value. The second step changes some instances of $\boldsymbol{z}_i$ into an integral over $\boldsymbol{v}$ which is only nonzero when $\boldsymbol{v}=\boldsymbol{z}_i$ (i.e. $\delta_{\boldsymbol{z}_i}$). Thereafter, the terms are rearranged such that the formula reduces to an expected value over $\boldsymbol{v}$ with weights proportional to the probability of generating $\boldsymbol{v}$ with the distribution $F$ and generating the query $\boldsymbol{u}$ with Gaussian noise $\mathcal{N}(\boldsymbol{0},\sqrt{d}\boldsymbol{I})$ added to $\boldsymbol{v}$. Scaling the variance of the multi-dimensional Gaussian noise by $\sqrt{d}$ reduces the impact of the dimensionality $d$ on the similarity $g(\boldsymbol{u};\boldsymbol{v},\sqrt{d}\boldsymbol{I})$ between $\boldsymbol{u}$ and $\boldsymbol{v}$.

The above derivation is inspired by the interpretation of softmax as Bayesian classification with normally distributed classes (Bishop, 1995), and it is similar to the interpretation of attention keys as latent mixture distributions provided by Nguyen et al. (2022). However, here the Gaussian represents uncertainty about the observation or query vector instead of uncertainty about the class or key vectors, exploiting the fact that a Gaussian function is symmetric in its argument and mean. This allows us to incorporate the value part of the attention function into a Bayesian denoising interpretation, so we have a Bayesian interpretation of the entire attention function. To the best of our knowledge, this interpretation of attention is novel.

The function from equation 14 is a special case of the definition of denoising attention $\mathrm{DAttn}(\boldsymbol{u}; F)$ given above in equation 3, where $F = F_{\boldsymbol{Z}}$. The construction above indicates that any scaled dot product attention function is an example of the $\mathrm{DAttn}(\boldsymbol{u}; F)$ function.[4] However, while attention is only defined over sets-of-vectors $\boldsymbol{Z}$, denoising-attention is defined over any probability distribution $F$ over a vector space, not just finite sets of impulse distributions.

## H DERIVING THE FACTORISED DIRICHLET PROCESS

In this section we derive an alternative factorisation of a DP which helps with the sampling method in Section 3.2. For notational convenience, in this section we use $c$ as the number of components for the base distribution instead of $n+1$. This is still intended to include both the output of the encoder and the prior component.

Here we provide the proof that
$$\mathrm{FDP}(\boldsymbol{G}^q, \boldsymbol{\alpha}^q) \quad = \quad \mathrm{DP}(G_0^q, \alpha_0^q)$$

where $\boldsymbol{G}^q = (G_1^q, ..., G_c^q)$, $\boldsymbol{\alpha}^q = (\alpha_1^q, ..., \alpha_c^q)$, $G_0^q = \sum_{i=1}^{c} \frac{\alpha_i^q}{\alpha_0^q} G_i^q$, $\alpha_0^q = \sum_{i=1}^{c} \alpha_i^q$, and $F \sim \mathrm{FDP}(\boldsymbol{G}^q, \boldsymbol{\alpha}^q)$ is defined as

$$
\begin{aligned}
F &= \sum_{i=1}^{c} \rho_i F_i \\
\boldsymbol{\rho} &\sim \mathrm{Dir}(\alpha_1^q, ..., \alpha_c^q) \\
F_i &\sim \mathrm{DP}(G_i^q, \alpha_i^q) \ \text{ for } i = 1, ..., c
\end{aligned}
$$

We start with the definition of a DP as an infinite symmetric Dirichlet distribution. A Dirichlet process $F \sim \mathrm{DP}(G_0, \alpha_0)$ can be defined as the limit of a sequence of finite Dirichlet distributions (see Teh (2010)):

$$
\begin{aligned}
F &= \sum_{k=1}^{\infty} \pi_k \delta_{\boldsymbol{z}_k} \\
\boldsymbol{\pi} &\sim \lim_{\kappa_0 \to \infty} \mathrm{Dir}(\frac{\alpha_0}{\kappa_0}, \overset{\kappa_0}{...}, \frac{\alpha_0}{\kappa_0}) \\
\boldsymbol{z}_k &\sim G_0 \ \text{ for } k = 1, ..., \infty
\end{aligned}
$$

Note that the weights $\boldsymbol{\pi}$ and the vectors $\boldsymbol{z}_k$ are independent of each other, so we can treat these two issues separately.

For the vectors, we know that after generating an infinite number of $\boldsymbol{z}_k$ from $G_0$, a proportion of exactly $\frac{\alpha_i^q}{\alpha_0^q}$ of them will be generated from $G_i^q$. For a finite number of vectors $\kappa_0$, let $\kappa_i$ be the number of $\boldsymbol{z}_k$ generated from $G_i^q$, for each $i$. So we have

$$\lim_{\kappa_0 \to \infty} \frac{\kappa_i}{\kappa_0} \quad = \quad \frac{\alpha_i^q}{\alpha_0^q}$$

Given the exchangeability of Dirichlet distributions, we can renumber the $\kappa_0$ categories of $\mathrm{Dir}(\frac{\alpha_0}{\kappa_0}, \overset{\kappa_0}{...}, \frac{\alpha_0}{\kappa_0})$ so that $\boldsymbol{\pi} = (\pi_{11}, ..., \pi_{1\kappa_1}, \overset{c}{...}, \pi_{c1}, ..., \pi_{c\kappa_c})$ and the $\pi_{i1}, ..., \pi_{i\kappa_i}$ are all weights for vectors $\boldsymbol{z}_{ij}$ generated from component $i$.

$$\boldsymbol{z}_{ij} \quad \sim \quad G_i^q \ \text{ for } i = 1, ..., c; \ j = 1, ..., \kappa_i$$

---

[4]There may be other constructions which could equally well be used to implement $\mathrm{Attn}(\boldsymbol{u}, \boldsymbol{Z})$ in terms of $\mathrm{DAttn}(\boldsymbol{u}; F)$, but all that is important here is the existence of one. Equally, we do not intend to claim that all $\mathrm{DAttn}(\boldsymbol{u}; F)$ functions can be implemented in terms of $\mathrm{Attn}(\boldsymbol{u}, \boldsymbol{Z})$. Indeed, the greater generality of $\mathrm{DAttn}(\boldsymbol{u}; F)$ is crucial in this work.

For the weights, we again consider the case of finite $\kappa_0$ before taking the limit as $\kappa_0$ goes to infinity, using the above indexing where categories $ij$ are partitioned according to their vector's base distribution component $i$. We define $(\rho_1, \overset{c}{...}, \rho_c)$ to be the vector of total weights $\rho_i = \sum_{j=1}^{\kappa_i} \pi_{ij}$ for each of these partitions $i$. By the rule for merging categories in a Dirichlet distribution, we know that these total weights are themselves distributed according to a Dirichlet distribution.

$$(\rho_1, \overset{c}{...}, \rho_c) \quad \sim \quad \mathrm{Dir}(\alpha_1^q, \overset{c}{...}, \alpha_c^q)$$

Now we take advantage of the neutrality property of Dirichlet distributions. It states that this vector $(\rho_1, \overset{c}{...}, \rho_c)$ of partition weights and all of the vectors $(\frac{\pi_{i1}}{\rho_i}, \overset{\kappa_i}{...}, \frac{\pi_{i\kappa_i}}{\rho_i})$ of normalised weights inside each partition are independent. In essence, this means that the only way that the weights inside each partition constrain each other is through normalisation, so when normalisation is factored out they become independent. This independence allows us to compute the distribution over $(\frac{\pi_{i1}}{\rho_i}, \overset{\kappa_i}{...}, \frac{\pi_{i\kappa_i}}{\rho_i})$ by simply marginalising out all the other categories. We first merge all the categories outside partition $i$ into a single category, whose weight is thus $1-\rho_i$. This gives us the marginalised Dirichlet distribution $(\pi_{i1}, \overset{\kappa_i}{...}, \pi_{i\kappa_i}, (1-\rho_i))$ $\sim \mathrm{Dir}(\frac{\alpha_0^q}{\kappa_0}, \overset{\kappa_i}{...}, \frac{\alpha_0^q}{\kappa_0}, \alpha_0^q(1-\frac{\kappa_i}{\kappa_0}))$. Let $d(\boldsymbol{\pi}; \boldsymbol{\alpha})$ be the probability density function for the distribution $Dir(\boldsymbol{\alpha})$:

$$d(\pi_{i1}, \overset{\kappa_i}{...}, \pi_{i\kappa_i}, (1-\rho_i); \frac{\alpha_0^q}{\kappa_0}, \overset{\kappa_i}{...}, \frac{\alpha_0^q}{\kappa_0}, \alpha_0^q(1-\frac{\kappa_i}{\kappa_0}))$$

$$= \frac{\Gamma(\alpha_0^q)}{\Gamma(\alpha_0^q(1-\frac{\kappa_i}{\kappa_0}))\prod_{j=1}^{\kappa_i}\Gamma(\frac{\alpha_0^q}{\kappa_0})}(1-\rho_i)^{\alpha_0^q(1-\frac{\kappa_i}{\kappa_0})-1}\prod_{j=1}^{\kappa_i}\pi_{ij}^{\frac{\alpha_0^q}{\kappa_0}-1}$$

$$= \frac{\Gamma(\alpha_0^q)}{\Gamma(\alpha_0^q(1-\frac{\kappa_i}{\kappa_0}))\prod_{j=1}^{\kappa_i}\Gamma(\frac{\alpha_0^q}{\kappa_0})}(1-\rho_i)^{\alpha_0^q(1-\frac{\kappa_i}{\kappa_0})-1}(\rho_i)^{\alpha_0^q\frac{\kappa_i}{\kappa_0}-1}(\prod_{j=1}^{\kappa_i}(\frac{\pi_{ij}}{\rho_i})^{\frac{\alpha_0^q}{\kappa_0}-1})$$

Now we can marginalise out the weight of the outside category by integrating over $\rho_i$.

$$\int_{\rho_i=0}^{1}\frac{\Gamma(\alpha_0^q)}{\Gamma(\alpha_0^q(1-\frac{\kappa_i}{\kappa_0}))\prod_{j=1}^{\kappa_i}\Gamma(\frac{\alpha_0^q}{\kappa_0})}(1-\rho_i)^{\alpha_0^q(1-\frac{\kappa_i}{\kappa_0})-1}(\rho_i)^{\alpha_0^q\frac{\kappa_i}{\kappa_0}-1}(\prod_{j=1}^{\kappa_i}(\frac{\pi_{ij}}{\rho_i})^{\frac{\alpha_0^q}{\kappa_0}-1})\,d\rho_i$$

$$= \left(\frac{\Gamma(\alpha_0^q)}{\Gamma(\alpha_0^q(1-\frac{\kappa_i}{\kappa_0}))\prod_{j=1}^{\kappa_i}\Gamma(\frac{\alpha_0^q}{\kappa_0})}\int_{\rho_i=0}^{1}(1-\rho_i)^{\alpha_0^q(1-\frac{\kappa_i}{\kappa_0})-1}(\rho_i)^{\alpha_0^q\frac{\kappa_i}{\kappa_0}-1}\,d\rho_i\right)(\prod_{j=1}^{\kappa_i}(\frac{\pi_{ij}}{\rho_i})^{\frac{\alpha_0^q}{\kappa_0}-1})$$

$$= d(\frac{\pi_{i1}}{\rho_i}, \overset{\kappa_i}{...}, \frac{\pi_{i\kappa_i}}{\rho_i}; \frac{\alpha_0^q}{\kappa_0}, \overset{\kappa_i}{...}, \frac{\alpha_0^q}{\kappa_0})$$

where in the last step we note that the integral (assuming it is well defined) is simply part of the normalisation constant, which we know from the definition of the Dirichlet distribution must be $B(\frac{\alpha_0^q}{\kappa_0}, \overset{\kappa_i}{...}, \frac{\alpha_0^q}{\kappa_0})$. This gives us

$$(\frac{\pi_{i1}}{\rho_i}, \overset{\kappa_i}{...}, \frac{\pi_{i\kappa_i}}{\rho_i}) \quad \sim \quad \mathrm{Dir}(\frac{\alpha_0^q}{\kappa_0}, \overset{\kappa_i}{...}, \frac{\alpha_0^q}{\kappa_0})$$

Now that we have all the individual distributions, we can put them together to get the factorised distribution for the case of finite $\kappa_0$.

$$\pi_{ij} \quad = \quad \rho_i\pi'_{ij} \text{ for } i=1,...,c; j=1,...,\kappa_i$$
$$\boldsymbol{\rho} \quad \sim \quad \mathrm{Dir}(\alpha_1^q, \overset{c}{...}, \alpha_c^q)$$
$$\boldsymbol{\pi}'_i \quad \sim \quad \mathrm{Dir}(\frac{\alpha_0^q}{\kappa_0}, \overset{\kappa_i}{...}, \frac{\alpha_0^q}{\kappa_0}) \text{ for } i=1,...,c$$

Noting that $\lim_{\kappa_0\to\infty}\frac{\alpha_0^q}{\kappa_0} = \frac{\alpha_i^q}{\kappa_i}$, we can then take the limit as $\kappa_0$ goes to infinity to get our definition of the weights for the factorised Dirichlet distribution.

$$\pi_{ij} \quad = \quad \rho_i\pi'_{ij} \text{ for } i=1,...,c; j=1,...,\infty$$
$$\boldsymbol{\rho} \quad \sim \quad \mathrm{Dir}(\alpha_1^q, \overset{c}{...}, \alpha_c^q)$$
$$\boldsymbol{\pi}'_i \quad \sim \quad \lim_{\kappa_i\to\infty}\mathrm{Dir}(\frac{\alpha_i^q}{\kappa_i}, \overset{\kappa_i}{...}, \frac{\alpha_i^q}{\kappa_i}) \text{ for } i=1,...,c$$

Thus the weights of a DP can be rewritten as the weights of an equivalent FDP using the above construction.

Putting the vectors and weights together, we get the distribution $F_i$ over the weighted vectors in each partition $i$.

$$
\begin{aligned}
\boldsymbol{z}_{ij} &\sim G_i^q \ \text{ for } i=1,...,c;\ j=1,...,\kappa_i \\
\boldsymbol{\pi}_i' &\sim \lim_{\kappa_i \to \infty} \mathrm{Dir}(\tfrac{\alpha_i^q}{\kappa_i},\overset{\kappa_i}{...},\tfrac{\alpha_i^q}{\kappa_i}) \ \text{ for } i=1,...,c
\end{aligned}
$$

and thus

$$
F_i \sim \mathrm{DP}(G_i^q,\alpha_i^q) \ \text{ for } i=1,...,c
$$

This concludes our proof that, if $F \sim \mathrm{DP}(G_0^q,\alpha_0^q)$, then:

$$
\begin{aligned}
F &= \sum_{i=1}^{c} \rho_i F_i \\
\boldsymbol{\rho} &\sim \mathrm{Dir}(\alpha_1^q,\overset{c}{...},\alpha_c^q) \\
F_i &\sim \mathrm{DP}(G_i^q,\alpha_i^q) \ \text{ for } i=1,...,c
\end{aligned}
$$

and thus $\mathrm{FDP}(\boldsymbol{G}^q,\boldsymbol{\alpha}^q)=\mathrm{DP}(G_0^q,\alpha_0^q)$.

## I  DERIVING THE KL DIVERGENCE

In this section we derive the KL divergence between the prior and the posterior. We also argue that this function is approximately linear in the number of sampled vectors for each component.

To directly compare the posterior with the prior, we first reformulate the prior as a factorised DP with the same form as the posterior. We can do this without changing the distribution specified by the prior, simply by making $n + 1$ copies of the base distribution $G_0^p$ and weighting those copies proportionately to the weights $\frac{\alpha_i^q}{\alpha_0^q}$ of the components of the posterior base distribution. This gives us the prior $\mathrm{BFDP}(\boldsymbol{G}^p,\boldsymbol{\alpha}^{p'},\kappa^\Delta)$ where $\boldsymbol{G}^p=(G_0^p,\overset{n+1}{...},G_0^p)$ and $\boldsymbol{\alpha}^{p'}=\boldsymbol{\alpha}^q\frac{\alpha_0^{p'}}{\alpha_0^q}=(\alpha_0^{p'}\frac{\alpha_1^q}{\alpha_0^q},\overset{n+1}{...},\alpha_0^{p'}\frac{\alpha_{n+1}^q}{\alpha_0^q})$.

The formulation of both the prior and posterior as bounded factorised DPs of the same form simplifies the computation of the KL divergence, because the KL divergence for each respective pair of factors can be computed separately, and then combined.

First consider the factors for the Dirichlet distributions over the partitions for the different components $i$. There is a closed-form solution to the KL divergence between two Dirichlet distributions.

$$
\begin{aligned}
&D_{\mathrm{KL}}\Big(\mathrm{Dir}(\alpha_1^q,\overset{n+1}{...},\alpha_{n+1}^q) \ \| \ \mathrm{Dir}(\alpha_0^{p'}\tfrac{\alpha_1^q}{\alpha_0^q},\overset{n+1}{...},\alpha_0^{p'}\tfrac{\alpha_{n+1}^q}{\alpha_0^q})\Big) \\
&= \ \log\frac{\Gamma(\alpha_0^q)}{\Gamma(\alpha_0^{p'})}+\sum_{i=1}^{n+1}\left(-\log\frac{\Gamma(\alpha_i^q)}{\Gamma(\alpha_0^{p'}\tfrac{\alpha_i^q}{\alpha_0^q})}+\alpha_i^q(1-\tfrac{\alpha_0^{p'}}{\alpha_0^q})(\psi(\alpha_i^q)-\psi(\alpha_0^q))\right)
\end{aligned}
$$

where $\Gamma$ is the gamma function and $\psi$ is the digamma function.

For the bounded DPs for each individual component $i$, there are two factors, a symmetric Dirichlet distribution over the weights and a Gaussian distribution over each vector. For the symmetric Dirichlet distribution, in the case where $\kappa_i = 1$, then the KL for this term is zero, since there is no choice to make for this weight. In the case where $\kappa_i > 1$, the KL divergence between the posterior and prior versions of these weight distributions again has a closed-form solution.

$$
\begin{aligned}
&D_{\mathrm{KL}}\Big(\mathrm{Dir}(\tfrac{\alpha_i^q}{\kappa_i},\overset{\kappa_i}{...},\tfrac{\alpha_i^q}{\kappa_i}) \ \| \ \mathrm{Dir}(\alpha_0^{p'}\tfrac{\alpha_i^q}{\alpha_0^q\kappa_i},\overset{\kappa_i}{...},\alpha_0^{p'}\tfrac{\alpha_i^q}{\alpha_0^q\kappa_i})\Big) \\
&= \ \log\frac{\Gamma(\alpha_i^q)}{\Gamma(\alpha_0^{p'}\tfrac{\alpha_i^q}{\alpha_0^q})}-\kappa_i\log\frac{\Gamma(\tfrac{\alpha_i^q}{\kappa_i})}{\Gamma(\alpha_0^{p'}\tfrac{\alpha_i^q}{\alpha_0^q\kappa_i})}+\alpha_i^q(1-\tfrac{\alpha_0^{p'}}{\alpha_0^q})(\psi(\tfrac{\alpha_i^q}{\kappa_i})-\psi(\alpha_i^q))
\end{aligned}
$$

This term then needs to be summed across components $1 \le i \le n+1$.

For the factors for generating vectors from each individual component of the base distribution, because the different components have been factorised, there is also a closed-form solution to computing these KL divergences. Each Gaussian component of the posterior's base distribution is compared independently to the Gaussian of the prior's base distribution. The KL divergence between two Gaussians (with diagonal

covariance with values $\boldsymbol{\sigma}$) is:

$$
\begin{aligned}
D_{\mathrm{KL}}(G_i^q \parallel G_0^p) &= \tfrac{1}{2}\sum_{h=1}^{d}\Big(\frac{(\mu_{ih}^q-\mu_h^p)^2}{(\sigma_h^p)^2}+\frac{(\sigma_{ih}^q)^2}{(\sigma_h^p)^2}-1-\log(\frac{(\sigma_{ih}^q)^2}{(\sigma_h^p)^2})\Big) \\
&= \tfrac{1}{2}\sum_{h=1}^{d}\big((\mu_{ih}^q)^2+(\sigma_{ih}^q)^2-1-\log((\sigma_{ih}^q)^2)\big)
\end{aligned}
$$

where the last step assumes $\boldsymbol{\mu}^p=\mathbf{0}$, $(\boldsymbol{\sigma}^p)^2=\mathbf{1}$. This term then needs to be multiplied by the number $\kappa_i$ of vectors for this component, and summed across components $1\le i\le n+1$.

Given these exact closed-form solutions for each pair of factors, we can compute the full KL divergence. We start by combining the formulas for the weight factors, where some terms cancel:

$$
\begin{aligned}
&D_{\mathrm{KL}}\Big(\mathrm{Dir}(\alpha_1^q,\overset{n+1}{\ldots},\alpha_{n+1}^q)\parallel \mathrm{Dir}(\alpha_0^{p'}\tfrac{\alpha_1^q}{\alpha_0^q},\overset{n+1}{\ldots},\alpha_0^{p'}\tfrac{\alpha_{n+1}^q}{\alpha_0^q})\Big) \\
&+\sum_{i=1}^{n+1}D_{\mathrm{KL}}\Big(\mathrm{Dir}(\tfrac{\alpha_i^q}{\kappa_i},\overset{\kappa_i}{\ldots},\tfrac{\alpha_i^q}{\kappa_i})\parallel \mathrm{Dir}(\alpha_0^{p'}\tfrac{\alpha_i^q}{\alpha_0^q\kappa_i},\overset{\kappa_i}{\ldots},\alpha_0^{p'}\tfrac{\alpha_i^q}{\alpha_0^q\kappa_i})\Big) \\
&= \log\frac{\Gamma(\alpha_0^q)}{\Gamma(\alpha_0^{p'})}-\sum_{i=1}^{n+1}\kappa_i\log\frac{\Gamma(\tfrac{\alpha_i^q}{\kappa_i})}{\Gamma(\tfrac{\alpha_0^{p'}\alpha_i^q}{\alpha_0^q\kappa_i})}+(1-\tfrac{\alpha_0^{p'}}{\alpha_0^q})\sum_{i=1}^{n+1}\alpha_i^q(\psi(\tfrac{\alpha_i^q}{\kappa_i})-\psi(\alpha_0^q))
\end{aligned}
$$

Now we can put all these pieces together.

$$
\begin{aligned}
&D_{\mathrm{KL}}(\mathrm{BFDP}(\boldsymbol{G}^q,\boldsymbol{\alpha}^q,\boldsymbol{\kappa})\parallel \mathrm{BFDP}(\boldsymbol{G}^p,\boldsymbol{\alpha}^q\tfrac{\alpha_0^{p'}}{\alpha_0^q},\boldsymbol{\kappa})) \\
&= \int_{\boldsymbol{\rho}}\!\int_{\boldsymbol{\pi}'}\!\int_{\boldsymbol{v}} d(\boldsymbol{\rho};\alpha_1^q,...,\alpha_{n+1}^q)\Bigg(\prod_{i=1}^{n+1}d(\boldsymbol{\pi}_i';\tfrac{\alpha_i^q}{\kappa_i},\overset{\kappa_i}{\ldots},\tfrac{\alpha_i^q}{\kappa_i})\prod_{i=1}^{n+1}\prod_{j=1}^{\kappa_i}G_i^q(\boldsymbol{z}_{ij})\Bigg) \\
&\qquad\qquad \log\frac{\mathrm{BFDP}(\boldsymbol{G}^p,\boldsymbol{\alpha}^q\tfrac{\alpha_0^{p'}}{\alpha_0^q},\boldsymbol{\kappa})}{\mathrm{BFDP}(\boldsymbol{G}^q,\boldsymbol{\alpha}^q,\boldsymbol{\kappa})}\; d\boldsymbol{\rho}\; d\boldsymbol{\pi}'\; d\boldsymbol{v} \\
&= D_{\mathrm{KL}}\Big(\mathrm{Dir}(\alpha_1^q,\overset{n+1}{\ldots},\alpha_{n+1}^q)\parallel \mathrm{Dir}(\alpha_0^{p'}\tfrac{\alpha_1^q}{\alpha_0^q},\overset{n+1}{\ldots},\alpha_0^{p'}\tfrac{\alpha_{n+1}^q}{\alpha_0^q})\Big) \\
&\quad+\sum_{i=1}^{n+1}D_{\mathrm{KL}}\Big(\mathrm{Dir}(\tfrac{\alpha_i^q}{\kappa_i},\overset{\kappa_i}{\ldots},\tfrac{\alpha_i^q}{\kappa_i})\parallel \mathrm{Dir}(\alpha_0^{p'}\tfrac{\alpha_i^q}{\alpha_0^q\kappa_i},\overset{\kappa_i}{\ldots},\alpha_0^{p'}\tfrac{\alpha_i^q}{\alpha_0^q\kappa_i})\Big)+\kappa_i D_{\mathrm{KL}}(G_i^q\parallel G_0^p) \\
&= \log\Gamma(\alpha_0^q)-\log\Gamma(\alpha_0^{p'})+\sum_{i=1}^{n+1}\kappa_i\Big(\log\Gamma(\tfrac{\alpha_0^{p'}\alpha_i^q}{\alpha_0^q\kappa_i})-\log\Gamma(\tfrac{\alpha_i^q}{\kappa_i})\Big)+(\alpha_0^q-\alpha_0^{p'})\Big(-\psi(\alpha_0^q)+\sum_{i=1}^{n+1}\tfrac{\alpha_i^q}{\alpha_0^q}\psi(\tfrac{\alpha_i^q}{\kappa_i})\Big) \\
&\quad+\tfrac{1}{2}\sum_{i=1}^{n+1}\kappa_i\sum_{h=1}^{d}\Big(\frac{(\mu_{ih}^q-\mu_h^p)^2}{(\sigma_h^p)^2}+\frac{(\sigma_{ih}^q)^2}{(\sigma_h^p)^2}-\log\frac{(\sigma_{ih}^q)^2}{(\sigma_h^p)^2}-1\Big) \qquad (15)
\end{aligned}
$$

Equation 15 gives the KL portion of the loss when we are given the full set $\boldsymbol{\kappa}$ of numbers of vectors $\kappa_i$ generated for each component $i$.

If we assume that the $\kappa_i$ are chosen stochastically, then we can take advantage of the fact that equation 15 is approximately linear in $\kappa_i$, when the variation in $\kappa_i$ is fairly small relative to the values of $\kappa_i$. The Gaussian term is exactly linear in $\kappa_i$, and the terms $\psi(\tfrac{\alpha_i^q}{\kappa_i})$ and $\kappa_i\Big(\log\Gamma(\tfrac{\alpha_i^q}{\alpha_0^q\kappa_i})-\log\Gamma(\tfrac{\alpha_i^q}{\kappa_i})\Big)$ are both approximately linear in $\kappa_i$. This allows us to approximate the expectation over $\kappa_i$ of this loss as this loss of the expectation over $\kappa_i$, as discussed in Section 3.1. In this case, this approximation of the full KL divergence is:

$$
\begin{aligned}
&D_{\mathrm{KL}}(\mathrm{BFDP}(\boldsymbol{G}^q,\boldsymbol{\alpha}^q,\boldsymbol{\kappa})\parallel \mathrm{BFDP}(\boldsymbol{G}^p,\boldsymbol{\alpha}^q\tfrac{\alpha_0^{p'}}{\alpha_0^q},\boldsymbol{\kappa})) \\
&\approx \log\Gamma(\alpha_0^q)-\log\Gamma(\alpha_0^{p'})+(\alpha_0^q-\alpha_0^{p'}))\Big(\psi(\tfrac{\alpha_0^q}{\kappa_0})-\psi(\alpha_0^q)\Big)+\kappa_0\Big(\log\Gamma(\tfrac{\alpha_0^{p'}}{\kappa_0})-\log\Gamma(\tfrac{\alpha_0^q}{\kappa_0})\Big) \quad (16) \\
&\quad+\tfrac{1}{2}\kappa_0\sum_{i=1}^{n+1}\tfrac{\alpha_i^q}{\alpha_0^q}\sum_{h=1}^{d}\Big(\frac{(\mu_{ih}^q-\mu_h^p)^2}{(\sigma_h^p)^2}+\frac{(\sigma_{ih}^q)^2}{(\sigma_h^p)^2}-1-\log\frac{(\sigma_{ih}^q)^2}{(\sigma_h^p)^2}\Big)
\end{aligned}
$$

## J    REPARAMETERISATION TRICK AND SAMPLING

In this section we consider the reparameterisation trick to allow backpropagation through the sampling step. We consider the component Gaussians of the base distribution and the weights generated by the DP separately.

### J.1    SAMPLING VECTORS FROM A COMPONENT OF THE BASE DISTRIBUTION

Each vector $z_k$ is sampled independently from some specific component $G_i^q$ of the base distribution $G_0^q$. Since we assume that all these components are distributed according to a Gaussian $G_i^q = \mathcal{N}(\boldsymbol{\mu}_i^q, \boldsymbol{I}(\boldsymbol{\sigma}_i^q)^2)$, we can sample from this distribution using location-scale shifting (Kingma & Welling, 2014):

$$
\begin{aligned}
z_k &= \boldsymbol{\mu}_i^q + \boldsymbol{\sigma}_i^q \epsilon_k \\
\epsilon_k &\sim \mathcal{N}(\boldsymbol{0}, \boldsymbol{1})
\end{aligned}
\tag{17}
$$

Since the random sampling comes from an unparameterised unit Gaussian, there is no need to backpropagate error into this sampling step, but the error can be backpropagated into $\boldsymbol{\mu}_i^q$ and $\boldsymbol{\sigma}_i^q$ given a specific sample. This is the reparameterisation trick for Gaussian distributions.

### J.2    SAMPLING FROM A DIRICHLET DISTRIBUTION

A Dirichlet distribution over category weights can be sampled by sampling from a Gamma distribution for each category and then normalising. A sum-normalised set of $\kappa$ random variables $\pi_1, ..., \pi_\kappa \sim Dir(\alpha_1, ..., \alpha_\kappa)$ follows a Dirichlet distribution if the unnormalised random variables $\gamma_i$ each follow a Gamma distribution.

$$
\begin{aligned}
\pi_i &= \frac{\gamma_i}{\sum_i^\kappa \gamma_i} \\
\gamma_i &\sim \Gamma(\alpha_i, \beta = 1)
\end{aligned}
\tag{18}
$$

where $\Gamma(\alpha_i, \beta = 1)$ is the Gamma distribution with $\beta = 1$, whose PDF is $f(x) = \frac{1}{\Gamma(\alpha_i)} \exp((\alpha_i - 1)\log(x) - x)$. There is no closed-form explicit reparameterisation trick for the Gamma distribution, but there are for approximations. We propose to use a combination of two approximations for the Gamma distribution which have a reparameterisation trick, one for small values of $\alpha_i$ and one for larger values of $\alpha_i$.[5]

**Inverse CDF approximation to Gamma distributions**    The Gamma distribution cannot use location-scale shifting for sampling due to its asymmetry, nor can the parameters and noise components be decoupled in the inverse CDF. Hence, (Knowles, 2015) suggests sampling using an approximation to the inverse CDF of the Gamma distribution of the following form:

$$
\begin{aligned}
\gamma_i &\approx \beta^{-1}(u_i \alpha_i \Gamma(\alpha_i))^{1/\alpha_i}, \\
u_i &\sim \mathrm{U}(0,1).
\end{aligned}
\tag{19}
$$

This approximation allows the inverse CDF of the Gamma distribution to be a function of the parameters and independent noise from a uniform distribution $\mathrm{U}(0,1)$. However, this approximation is only recommended when the value of $\alpha_i < 1$ and $\beta = 1$. In our case $\beta = 1$ but we sometimes have large $\alpha_i$.

**Gaussian approximation to Gamma distributions**    Knowles (2015) further mentions that the Gaussian distribution can be used to approximate a Gamma distribution for larger $\alpha$. Bahuleyan et al. (2018) uses a similar approach to approximate variational attention weights. The Gaussian distribution is a symmetric distribution which can be sampled by location-scale shifting, as discussed above. The Gamma distribution $\Gamma(\alpha, \beta = 1)$ can be approximated with a Gaussian of the form $\gamma \sim \mathcal{N}(\alpha, \sqrt{\alpha})$ (Knowles, 2015), which gives us:

$$
\begin{aligned}
\gamma_i &\approx \alpha_i + \sqrt{\alpha_i} \epsilon_i \\
\epsilon_i &\sim \mathcal{N}(0,1)
\end{aligned}
\tag{20}
$$

The Gaussian distribution is symmetric and can take on negative values. Hence, this approximation is inappropriate for the Gamma distribution unless the $\alpha_i$ parameter is sufficiently large, otherwise the sample will need to be truncated to a value greater than zero.

---

[5]We leave the investigation of implicit reparameterisation gradients (Figurnov et al., 2018) to future work. This approach is an alternative to explicit reparameterisation for cases like Gamma distributions, but first we investigate the approach which is more standard in the VAE literature.

**The combined reparameterisation of Gamma distributions**   To visualise the error for these two approximations, their average $L_1$ distance from the true Gamma inverse CDF is plotted in Figure 9 for different values of $\alpha$. The plot shows that the approximation error is equal when $\alpha = 0.6363$. To take advantage of the strengths of both these approximations, we propose to reparameterise the Gamma distribution as a blend of these two approximations with a switch at $\alpha = 0.6363$ and truncate negative samples to zero.

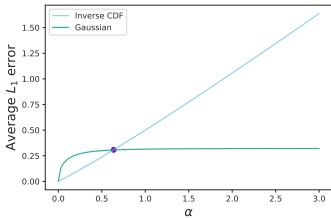

Figure 9: The average absolute difference between a Gamma inverse CDF function and our approximations: inverse CDF approximation and Gaussian inverse CDF are plotted for values of $\alpha$.

## K   PRACTICAL IMPLEMENTATION OF DENOISING ATTENTION

In this section we provide the equations used to allow denoising attention to be implemented in a deep learning framework at training time and test time.

**Denoising attention at training time**   During training, the set of vectors $\boldsymbol{Z} \in \mathbb{R}^{n \times p}$ and their log-probability values $\log(\boldsymbol{\pi}) \in \mathbb{R}^{1 \times n}$ are both sampled and output by the NVIB layer, thereby specifying the sampled mixture distribution $F$. For each use of denoising attention, the query $\boldsymbol{u}' \in \mathbb{R}^{1 \times p}$ is projected by the grouped matrices $\boldsymbol{W}^Q, \boldsymbol{W}^K \in \mathbb{R}^{p \times d}$ to $\boldsymbol{u} = (\boldsymbol{u}' \boldsymbol{W}^Q (\boldsymbol{W}^K)^T)$. The keys' dimensionality $d$ is used for scaling. Denoising attention can then be computed as:

$$\mathrm{DAttn}(\boldsymbol{u}; F) \quad = \quad \mathrm{softmax}\left(\tfrac{1}{\sqrt{d}}\boldsymbol{u}\boldsymbol{Z}^T + \log(\boldsymbol{\pi}) - \tfrac{1}{2\sqrt{d}}\|\boldsymbol{Z}\|^2\right)\boldsymbol{Z}$$

We define this for $\boldsymbol{u} \in \mathbb{R}^{1 \times p}$, but this can easily be extended to multiple queries.

**Denoising attention at test time**   During test time, we do not sample $F$, but instead use the mean of the posterior distribution, which is its base distribution $G_0^q$. The NVIB layer takes its input from the encoder and maps it to the parameters $(\boldsymbol{\mu}^q, \boldsymbol{\sigma}^q, \frac{\boldsymbol{\alpha}^q}{\alpha_0^q})$ of this base distribution $G_0^q = \sum_i \frac{\alpha_i^q}{\alpha_0^q}\mathcal{N}(\boldsymbol{\mu}_i^q, \boldsymbol{I}(\boldsymbol{\sigma}_i^q)^2)$. For convenience let $(\boldsymbol{\sigma}_i^r)^2 = (\sqrt{d} + (\boldsymbol{\sigma}_i^q)^2)$. Test-time denoising attention can then be computed as:

$DAttn(\boldsymbol{u}; G_0^q)$

$$= \ \mathrm{softmax}\left(\boldsymbol{u}\left(\frac{\boldsymbol{\mu}^q}{(\boldsymbol{\sigma}^r)^2}\right)^T + \log(\frac{\boldsymbol{\alpha}^q}{\alpha_0^q}) - \left(\frac{1}{2}\left\|\frac{\boldsymbol{\mu}^q}{\boldsymbol{\sigma}^r}\right\|^2\right)^T - \boldsymbol{1}_p(\log(\boldsymbol{\sigma}^r))^T\right)\left(\frac{(\boldsymbol{\sigma}^q)^2}{(\boldsymbol{\sigma}^r)^2}\odot(\boldsymbol{1}_n^T\boldsymbol{u}) + \frac{\sqrt{d}}{(\boldsymbol{\sigma}^r)^2}\odot\boldsymbol{\mu}^q\right)$$

where $\boldsymbol{1}_p$ is a row vector of $p$ ones.

A caveat of this derivation is that it applies only for single-head attention and is not trivial to extend for multi-head attention. We leave this for future work.

