# OpenReview forum: "A VAE for Transformers with Nonparametric Variational Information Bottleneck"
_ICLR.cc/2023/Conference — ICLR 2023 poster_

### Official Review · Reviewer_fw7U · 2022-10-23

**Confidence:** 4
**Correctness:** 3
**Technical Novelty And Significance:** 2
**Empirical Novelty And Significance:** 2
**Recommendation:** 5

**Clarity, Quality, Novelty And Reproducibility:**

Clarity:

The general clarity of the paper is OK. But I do have confusions on connecting the ELBO with the attention architecture of the transformer. The ELBO shown in Section 3.1 is a general form of gaussian mixture. It is unclear to me that how to learn the parameters of attention by maximizing the ELBO. Clearer formulations of p(x|F) and q(F|x) with the attention parameters need to be introduced.

**Strength And Weaknesses:**

Strength:

1. The (re) formulation of attention to a gaussian mixture and the application of variational inference are interesting and intuitive.

2. The paper provides comprehensive experiments on parameter sensitivity and ablation study.

Feedbacks (not all about weakness)

1. Although the formulation of a gaussian mixture of attention is interesting and contributes to the significance of the paper, I think the technical depth is not large enough for an accept yet. Using variational inference or VAE with gaussian mixtures is a well studied technique, which is directly applied given the formulation.

2. I don't see it is necessary to highlight Bayesian nonparametric or Dirichlet processes. First of all, DPs are usually applied into latent variables so that the dimensions of them can grow with data. In the case of this paper, the dimension of Z is the batch size of data, which does not usually grow in training or testing. Secondly, although discussing Bayesian nonparametric or Dirichlet processes, the paper ends up with a vanilla Dirichlet gaussian mixture model, which does not have strong connections with Dirichlet processes. It is not even a truncated approximation (e.g. stick breaking process) of DP. The paper names this unbounded DP and claims it as an approximation of DP, which requires theoretical justification.

3. In the experiments, the paper mainly compares with the baselines of different configurations of VAE with transformers. I think it is expected to put the proposed method in the category of text generation methods and compare it with other kinds of methods in the same category. This is to show how the method is positioned in the task.

**Summary Of The Paper:**

This paper proposes a VAE framework for that formulates attentions in transformers as a mixture of gaussians, where variational inference can be used to learn the framework. The proposed VAE with transformers as the encoder and decoder is used in text reconstruction and generation tasks.

**Summary Of The Review:**

The idea of the paper is interesting but I feel the technical depth is not enougth.

---

> ### Author Response · Authors · 2022-11-14
> **Misunderstandings of our contributions and their difference from previous uses of nonparametrics.**
>
> We are pleased that reviewer fw7U found the paper interesting, intuitive and comprehensive. However, we believe the
> reviewer has misunderstood our contributions. We have made the introduction of the paper clearer, and address specific
> misunderstanding here. The paper’s technical innovations are much greater than what the reviewer thinks we are proposing.
>
> Strength 1: We do not use Gaussian mixtures in our interpretation of attention. We use mixtures of impulses with Gaussian
> noise around the query.
>
> Feedback 1: We are not proposing to use Gaussian mixtures over latent vectors, which has indeed been well studied. We are
> proposing to use mixture distributions as the latent space itself, and use Dirichlet Processes to define distributions over these
> latent mixture distributions.
>
> Feedback 2, first half: The dimension of $Z$ is not the batch size of the data, but the sequence length of the input text, which
> varies across individual examples. It is indeed common to use Bayesian nonparametrics to increase the size of the model
> depending on the amount of training data, but that is not what we are doing. We use Bayesian nonparametrics so that, at test
> time, the size of the latent representation can change depending on the size of the individual input text. To the best of our
> knowledge, this is a novel use of Bayesian nonparametrics which has never previously been used to define a VAE.
>
> Feedback 2, second half: Although we do approximate a Dirichlet process with a Dirichlet distribution, the parameterisation is
> the same as a Dirichlet process. This is what allows us to change the degree of approximation (i.e. the number of categories) at
> test time without needing to learn new parameters. It is an approximation in that it converges to an exact DP as the number of
> categories goes to infinity (Teh, 2010). Please see Section 2.2.
>
> Feedback 3: We do not claim improvements in the SOTA on text generation tasks, since this would require an entire paper on
> its own. Our experiments are a proof-of-concept to demonstrate that our theoretical results work in practice, and that there is
> strong potential for future work on this topic.
>
> Clarity: The confusion around the role of the ELBO (section 3.1) has its roots in the above misunderstandings of earlier
> sections. We have improved the linking between this section and earlier sections. We are not learning the parameters of the
> attention function. We are learning an encoder function $q(F |x)$ from the input text $x$ to a distribution over mixtures of impulse
> distributions $F $, one of which will be an input to the cross attention function of the decoder $p(x|F )$ to reconstruct the input.
>
> We hope that with these explanations and the revised submission, reviewer fw7U will find this paper’s innovative approach
> and numerous technical contributions as exciting as we do.

---

### Official Review · Reviewer_tsdp · 2022-10-24

**Confidence:** 3
**Clarity, Quality, Novelty And Reproducibility:** Paper needs more work in finalization.
**Correctness:** 4
**Technical Novelty And Significance:** 3
**Empirical Novelty And Significance:** 3
**Recommendation:** 6

**Strength And Weaknesses:**

Strengths:
- Novel probabilistic formulation of attention model in transformers.
- Full derivation is shown.
- Model has potential usefulness beyond the experiments shown in the paper.

Weaknesses:
- Implicit reparametrization, that provides exact reparametrization for Dirichlet is not used ( https://arxiv.org/abs/1805.08498)
- Manuscript seems to be typeset in a hurry, for example p(F|x) and p(F) are both named as prior. AFAIK, prior should not have data-term conditioning. Please clarify this point.
- Even though mathematical development is convincing, authors have not motivated their work. They should clearly explain in Introduction that for what reason this development was done. It cannot be that just this formulation has never been done before!

**Summary Of The Paper:**

In this manuscript authors embedding space of Transformer encoders as mixture distributions and as such are able to formulate transformer encoder - decoder as a VAE model. Number of mixture distributions is variable to authors decide to use Bayesian nonparametrics to solve the modeling issue, namely they use bounded Dirichlet prior.

**Summary Of The Review:**

All in all, I find the paper quite interesting and potentially very useful. I think the way how transformer is reformulated as a VAE model is quite neat. More large-scale experiments obviously would make the paper much better as authors also state in the Conclusions.

---

> ### Author Response · Authors · 2022-11-14
> **Thank you for the helpful suggestions.**
>
> We would like to the thank reviewer tsdp for their thoughtful review and encouraging assessment of the novelty, interest,
> and potential usefulness of our contributions. We agree with all the reviewer’s comments and we have improved the revised
> version accordingly.
>
> Weakness 1: We thank the reviewer for their helpful suggestion to use implicit reparametrization for our distributions. It
> is not clear whether it will work better than the approximate reparameterisation we propose, but we will definitely evaluate
> this in future work.
>
> Weakness 2: The reviewer is correct that there was a mistake in wording in Section 2.2 when referring to $p(F |x)$, which
> is in fact the true posterior, in contrast with its approximation $q(F |x)$. We also discuss a "conditional prior", but this only
> sees the length of the input $x$, and not $x$ itself. We have corrected the paper accordingly.
>
> Weakness 3: With regard to the motivation, please see our general response. We believe that the revised introduction addresses
> the issue of motivation more clearly and explicitly.

---

### Official Review · Reviewer_w4Qq · 2022-10-25

**Confidence:** 3
**Correctness:** 4
**Technical Novelty And Significance:** 4
**Empirical Novelty And Significance:** 4
**Recommendation:** 6

**Clarity, Quality, Novelty And Reproducibility:**

Clarity:
The grammar is kind of strangely worded, but overall fine. There is a lot of content, but sometimes key details seems to be glossed over (see above box).

**Strength And Weaknesses:**

Strengths:
- The paper seems interesting and a rather novel concept. Their approach is sound and the results seem ok. There are no glaring flaws in the paper.

Weaknesses:
- The point / takeaways / motivations are not very clear. Is the main benefit the regularization ability and the computational savings of having a lower dimensional latent space?
- While transfomer based models e.g. BERT / GPT-2 have been shown to be very useful, the paper is showing results on a a two layer Transformer encoder and decoder with a single attention-head. While this is the original proposed structure for using attention, it does not mimic either the encoder stack models e.g. BERT or the decoder stack models e.g. GPT-2 that are now the SOTA.
- Furthermore, it is not clear whether learning a NVAE for each attention mechanism (BERT would have 12*12=144 head) is computationally feasible and whether this training process is significantly more expensive than the standard learning of the query, key, and value weights.
- In the experimental results, the NVAE model is compared against VTP and VT (which don't seem to have any hyperparameters as only 1 best model was chosen?) and a hand-coded solution VTS. The performance of NVAE and VTS seems relatively comparable, but is VTS supposed to be an "oracle" or is there any further description on what "hand coded" means?
- In tables 7 and 8, all generated samples seem equally senseless? Similarly the interpolation examples (tables 9-12) seem to show all models are comparable?


**Summary Of The Paper:**

This paper presents a probabilistic mixture model representation of the attention mechanism in transformers. The probabilistic representation  allows the authors to define priors and posteriors over the mixture distributions and ultimately redefine the transformer as a nonparametric variational autoencoder. They show experimental results for the reconstruction of input sentences, generation by sampling from the prior, regularize the size of the latent space, and interpolate.

**Summary Of The Review:**

The paper is promising although it could be reworked to be stronger with a more clear focus and motivation.

---

> ### Author Response · Authors · 2022-11-14
> **Experiments, computational properties, and scope.**
>
> We would like to thank the reviever w4Qq for their thorough consideration of our experiments. We are motivated by the
> positive affirmation that our approach is sound, interesting and a novel concept. We are happy that the reviewer saw that
> computational savings are a potential benefit of our sparse latent representations, although we did not have space to investigate
> this issue in this paper. Please see the general response regarding the motivation and other benefits of our methods (Weakness
> 1). Specific questions are answered below.
>
> Experiments: (Weakness 2) Our experiments are designed as a proof-of-concept to show that our theoretical proposals
> work in practice. The relative simplicity of our Transformer encoders and decoders is primarily to stay within our limited
> computational budget. We provide promising larger scale experiments within our reach in Ablation B.4. We would expect
> that the effectiveness of our NVIB regulariser would become even more important as the size and power of the encoder and
> decoder increase to the scale of BERT or GPT-2 or larger, due to the over-parameterisation of these large models.
>
> (Weakness 5) In the absence of pretraining and large scale experiments, the decoded outputs show less human readable
> examples. Nevertheless, the metrics F-PPL and R-PPL (Zhao et al., 2017; Cífka et al., 2018) are used to measure the closeness
> of the distribution of generated text to the data where our model is better than the baselines.
>
> Computation: (Weakness 3) The possibility of extending our proposed NVIB layer to the self-attention layers is a very
> exciting direction of future work. As the reviewer points out, computational considerations will probably be important in
> doing this, but note that the only additional computation during training is the KL calculation and adding a bias to the attention
> weights (see Appendix Section J). For our current experiments the difference in computation time was negligible, but a
> full computation analysis was beyond the scope and claims of this paper. Also note that this extension could be done on
> only a limited number of layers, using faster versions of self-attention to do the bulk of the computation but still regularising
> intermediate representations. This could even have computational advantages due to the sparsity properties of NVIB, as
> the reviewer noticed. We plan to explore this in future work.
>
> Baselines: (Weakness 4) All our baselines underwent a hyperparameter search for their hyperparameters and were evaluated
> across seeds (Appendix Section B). Our baseline VTP is a typical VAE with a fixed latent dimension, and VT is a VAE
> with a variable latent dimension (i.e. the size of the input sequence). We see in Section 5.1 Figure 2a,b,c that these models
> either fail in generation (R-PPL or F-PPL) or reconstruction (BLEU). We provide a bridge between these baselines with
> the strided models (VTS). These models "hand code" the the proportion of vectors by dropping every second (third, forth,
> ... etc) latent vector based on its position in the sentence, and not learned based on information or distribution of the data.
>
> (Clarity) This paper covers a huge amount of material, and we had to make hard choices about what details to move into the
> appendix. The revised version clarifies the above points. We believe that we have made the body of the paper self-contained,
> allowing people with more specialised interests can consult the appendix.

---

### Official Review · Reviewer_Lu5K · 2022-10-31

**Confidence:** 3
**Correctness:** 3
**Technical Novelty And Significance:** 3
**Empirical Novelty And Significance:** Not applicable
**Recommendation:** 5

**Clarity, Quality, Novelty And Reproducibility:**

This paper is very detailed. The author included dozens of experiment setups and math in the supplement material, which makes their claims convincing and reproducible.

The motivation for this paper is not clear to me. For example, in the first paragraph of the Introduction, I am not clear why we should combine the transformer and VAE. If the motivation is to combine the strength of transformers and VAEs, which aspect of VAE could be improved in theory?

**Strength And Weaknesses:**

My biggest concern is in the experiment part.

First, maybe I am wrong, but in my opinion, this work may be compared with SOTA works to support their claim. I am not convinced at least at the current reviewing stage. I am not convinced why we should choose NVAE rather than the original VAE. Because, the VAE could also do the reconstruction, generation, interpolation, etc. It is clear to me the VAE is more complex compared with VAE, with more inductive bias, but it is not clear to me NVAE whether NVAE could reach a significant improvement in performance or could do something VAE couldn't.

Second, the prior is a mixture distribution, could different distributions learn different aspects of the input object? e.g. a disentangled latent representation? Some ablation studies or visualizations may be needed.

**Summary Of The Paper:**

This paper mainly proposed the nonparametric variational autoencoder (NVAE) by 'combining' VAEs and the transformer.
Further, the transformer encoder and decoder are used in the VAEs framework.
Different from VAEs whose prior is the standard Gaussian distribution, the proposed NVAE applied the nonparametric variational information bottleneck (NVIB) regulariser for the latent embedding.
Finally, the experiment shows NVAE could do reconstruction, generation, and regularisation tasks. The interpolation further explored the meaningfulness of the learned representation.

**Summary Of The Review:**

As above, my main concern is about the experiment, and the second is the motivation.

---

> ### Author Response · Authors · 2022-11-14
> **Improvements over a fixed-length-vector bottleneck, baselines and scope.**
>
> We extend our gratitude to reviewer Lu5K for their time and we are happy that they consider our paper to be thorough
> with respects to our claims, experiments and mathematical derivations. Concerning improvements to the motivation, please
> consider our general response as well as the discussion below.
>
> Regarding the advantages of NVAE over VAE, we should note that our baseline VTP represents a good example of a standard
> vector-space VAE applied to a standard text corpus. The empirical comparison of VTP to NVAE in Section 5.1, Figures
> 2a,b, shows that this vector-space VAE is not even able to learn to reconstruct accurately, as well as having poor generation
> diversity. The basic problem is that a fixed-length-vector embedding doesn’t have enough capacity (over-regularised) for
> long sentences, while having too much capacity (under-regularised) for short sentences. NVAE solves this problem, resulting
> in good reconstruction and generation metrics. Our other baselines (VT, VTS) represent hand-coded solutions to this problem,
> where the capacity grows linearly with the sentence length but it is not learned.
>
> Further discussion of the motivation for extending VAEs is given in the general response. We welcome the suggestion that
> we need to improve our explanation of this issue in the revised paper, which we have done.
>
> Regarding adding a comparison of our models to the SOTA and a disentanglement analysis, both these important topics
> must be left for future work. Our contributions are primarily focused on the theory of deep variational Bayesian models,
> so we provide empirical results only as a proof-of-concept. The paper is already extremely dense, so adding such complex
> topics within the page limit would be impossible. In addition, a meaningful comparison to the SOTA would require large-scale
> computational resources which are beyond our current budget. Both these issues are topics of current work, but they cannot
> be published until after the publication of the work in this submitted paper.

---

### Author Response · Authors · 2022-11-14
**General response.  Motivations and novelty.**

We thank all the reviewers for their effort and appreciate all the feedback received. We are pleased that reviewers found our
work novel (w4Qq, tsdp), interesting (w4Qq, tsdp, fw7U), comprehensive with respects to the mathematics (Lu5K, tsdp)
and experiments (Lu5K, fw7U) and with broad application (tsdp).

Three of the reviewers (Lu5K, w4Qq, tsdp) have asked for more motivation for our work, which we agree was understated. At
the risk of coming across as polemical, we argue for the importance and novelty of this work here. Our technical contributions
have made it possible to combine the advantages of attention-based models with the advantages of VAEs, which can be
motivation either as extending Transformers (w4Qq, tsdp) or as extending vector-space VAEs (Lu5k).

Extending Transformers: There is a large literature on the advantages of VAE (Mathieu et al., 2019; Ghosh et al., 2020;
Vahdat and Kautz, 2020) and VIB (Alemi et al., 2019) models. These advantages can be summarised as regularisation,
generation, and disentanglement. All these properties are important and active areas of research in NLP and are important
for improving Transformers. Regularisation and sparcity (w4Qq) has become increasingly more important for efficiency
and generalisation of large often over-parameterised language models (Child et al., 2019; Paranjape et al., 2020; Mahabadi
et al., 2021). Generation is a desirable property (fw7U) and a very active area of research in NLP (Liu and Liu, 2019; Brown
et al., 2020; Gong et al., 2022). Finally, disentanglement in NLP is an important topic (Lu5k) to understand the structure
of language and enhance explainability (Huang et al., 2021; Mercatali and Freitas, 2021; Correia et al., 2019). Our model
shows potential in all these sectors with each property being enough content for an individual paper. In this work we make
a theoretical proposal of how to extend Transformers with VIB, and report proof-of-concept experiments which focus on
the core properties of regularisation (Sections 5.2 and 5.3) and general generation abilities (Section 5.1).

Extending VAEs: Before the introduction of attention (Bahdanau et al., 2015), deep learning methods for NLP, such as
neural machine translation, used fixed-length vectors as their text embeddings. These latent representations have been almost
completely replaced by set-of-vector representations accessed with attention, with one vector per input token, such as the
output of a Transformer encoder. Despite this success, currently even VAEs built from Transformers still have fixed-length
vectors as their latent representations (Liu and Liu, 2019; Wang and Wan, 2019; Lin et al., 2020; Fan et al., 2020; Fang
et al., 2021; Zhao et al., 2021). Allowing VAEs to take advantage of attention-based latent representations is the problem
we address in this paper, thereby bringing VAE research in line with the progress in the rest of NLP. We propose a VAE
where the encoder outputs (a generalisation of) a set of vectors, and the decoder accesses this embedding with (a generalisation
of) cross attention. Our proposed VIB layer for this VAE regularises the (effective) number of vectors in the set, as well
as the information conveyed by each vector.

Novelty: This advance is achieved with a novel variational Bayesian approach to such attention-based representations, using
Bayesian nonparametrics. Bayesian nonparametrics has been extensively studied to allow the size of a model to grow with
the amount of training data, and has even been proposed for allowing the size of the latent vector space to grow with the
complexity of the examples found in the training data (Nalisnick and Smyth, 2016a;b). But, to the best of our knowledge,
it has never previously been proposed as a way to vary the size of an individual embedding dynamically at test time depending
on the arbitrarily large complexity of the individual example being embedded, as we do here. In addition, we derive a novel
interpretation of attention over a set of vectors as Bayesian query denoising with a mixture of impulse distributions prior.
This provides deep insights into Transformers and supports variational Bayesian approaches to such models, including our
distributions over attention-based representations with Dirichlet Processes.

To the extent that space permits, we have incorporated this argument in the introduction.

---

### Comment · Area_Chair_WXXq · 2022-11-15
**Please engage before the author-reviewer discussion closes**

Dear authors and reviewers,

The first phase of the discussion period is about to close on November 18.

For authors, please make sure to submit your rebuttal by the deadline. Leave some time for the reviewers to read it and respond while you are still allowed to further engage with them. Interactions between authors and reviewers are very important for the quality of the review process, so please make sure to engage.

For reviewers, please try to acknowledge and respond to the authors' rebuttal while the discussion period is still open for them to further interact with you.

Thank you for your participation in the review process!

Best,
The AC

---

### Decision · Program_Chairs · 2023-01-20

**Decision:**

Accept: poster

**Justification For Why Not Higher Score:**

Experiments remain a proof-of-concept.

**Justification For Why Not Lower Score:**

Large-scale experiments would require extensively more computation. I believe the current paper already makes an interesting contribution to VAEs and Transformers. Other concerns of the reviewers have been addressed.

**Metareview: Summary, Strengths And Weaknesses:**

The paper has received borderline reviews (5-6-6-5). The reviewers expressed concerns about the experimental validation of the proposed extension and would have liked to see how it performs on larger transformer architectures. They would also have appreciated results against state-of-the-art models for text generation tasks. The authors argue that the experiments are rather a proof-of-concept that only demonstrates the validity of the theoretical developments proposed in the paper (i.e., the development of a non-parametric variational information bottleneck for attention-based representations, and the development of the proposed Transformer VAE using NVIB).

As an area chair, I believe the authors have properly argued their case and made an interesting contribution to both VAEs and Transformers. For these reasons, I tend to recommend accepting the paper.

**Note From Pc:**

if the above contains the word "oral" or "spotlight" please see: "oral" presentation means -> notable-top-5% and "spotlight" means -> notable-top-25%. As stated in our emails, we are disassociating presentation type from AC recommendations

**Summary Of Ac-Reviewer Meeting:**

The paper will be discussed with the SAC.